# Inhibiting EZH2 targets atypical teratoid rhabdoid tumor by triggering viral mimicry via both RNA and DNA sensing pathways

Shengrui Feng [1,2,14] ✉, Sajid A. Marhon [1,14], Dustin J. Sokolowski[3,4,14], Alister D'Costa[5,6], Fraser Soares [1], Parinaz Mehdipour [1,7], Charles Ishak[1], Helen Loo Yau [1,8], Ilias Ettayebi[1,8], Parasvi S. Patel[1,8], Raymond Chen[1,8], Jiming Liu[9], Philip C. Zuzarte[6], King Ching Ho[10,11], Ben Ho[12], Shiyao Ning [1], Annie Huang[2,10,11,12], Cheryl H. Arrowsmith [1,8,13], Michael D. Wilson [3,4], Jared T. Simpson[5,6] & Daniel D. De Carvalho [1,8] ✉

Inactivating mutations in *SMARCB1* confer an oncogenic dependency on EZH2 in atypical teratoid rhabdoid tumors (ATRTs), but the underlying mechanism has not been fully elucidated. We found that the sensitivity of ATRTs to EZH2 inhibition (EZH2i) is associated with the viral mimicry response. Unlike other epigenetic therapies targeting transcriptional repressors, EZH2i-induced viral mimicry is not triggered by cryptic transcription of endogenous retro-elements, but rather mediated by increased expression of genes enriched for intronic inverted-repeat Alu (IR-Alu) elements. Interestingly, interferon-stimulated genes (ISGs) are highly enriched for dsRNA-forming intronic IR-Alu elements, suggesting a feedforward loop whereby these activated ISGs may reinforce dsRNA formation and viral mimicry. EZH2i also upregulates the expression of full-length LINE-1s, leading to genomic instability and cGAS/STING signaling in a process dependent on reverse transcriptase activity. Co-depletion of dsRNA sensing and cytoplasmic DNA sensing completely rescues the viral mimicry response to EZH2i in SMARCB1-deficient tumors.

The SWI/SNF chromatin remodeling complexes are important tumor suppressors with multiple subunits of SWI/SNF chromatin remodeling complexes inactivated by mutations in 20% of all cancers[1]. Specifically, almost all malignant rhabdoid tumors (MRTs) and ATRTs, which are highly aggressive childhood cancers without effective therapy, contain biallelic inactivating mutations in the SMARCB1 (SNF5, INI1, BAF47)

subunit[2,3]. The loss of SMARCB1 results in aberrant nucleosomal positioning by the SWI/SNF complex and elevated expression of EZH2[4,5]. EZH2 is the catalytic subunit of the Polycomb repressive complex 2 (PRC2), a complex that maintains gene repression by depositing the epigenetic silencing mark histone 3 lysine 27 tri-methylation (H3K27me3) on chromatin[6]. Extensive studies have

[1]Princess Margaret Cancer Centre, University Health Network, Toronto, ON, Canada. [2]The First Affiliated Hospital of University of South China, Hengyang, Hunan, China. [3]Department of Molecular Genetics, University of Toronto, Toronto, ON, Canada. [4]Genetics and Genome Biology, SickKids Research Institute, Toronto, ON, Canada. [5]Department of Computer Science, University of Toronto, Toronto, ON, Canada. [6]Ontario Institute for Cancer Research, Toronto, ON, Canada. [7]Ludwig Institute for Cancer Research, Nuffield Department of Medicine, University of Oxford, Oxford, UK. [8]Department of Medical Biophysics, University of Toronto, Toronto, ON, Canada. [9]The Cardiac Development and Early Intervention Unit, West China Institute of Women and Children's Health, West China Second University Hospital, Sichuan University, Chengdu, China. [10]Division of Hematology/Oncology, Hospital for Sick Children, Toronto, ON, Canada. [11]Arthur and Sonia Labatt Brain Tumour Research Centre, Hospital for Sick Children, Toronto, ON, Canada. [12]Laboratory Medicine and Pathobiology, Faculty of Medicine, University of Toronto, Toronto, ON, Canada. [13]Structural Genomics Consortium, University of Toronto, Toronto, ON, Canada. [14]These authors contributed equally: Shengrui Feng, Sajid A. Marhon, Dustin J. Sokolowski. ✉e-mail: shengrui.feng@uhn.ca; daniel.decarvalho@uhnresearch.ca

revealed that EZH2 is overexpressed in various malignancies[7–11] and acquires gain-of-function mutations in subtypes of lymphomas[12–14], highlighting an oncogenic role for EZH2 in cancer progression. Targeting EZH2 in cancers with activating EZH2 mutations or overexpression of EZH2 diminishes tumor growth, shown in preclinical studies[15–17]. Of note, many studies have uncovered a shared dependency of cancers with mutant SWI/SNF subunits on EZH2 activity[4,5,18–20]. Since almost all ATRTs are characterized by biallelic mutations of *SMARCB1*, these tumors may develop dependency on EZH2 that can be targeted pharmacologically through a synthetic lethality approach[17,21]. Clinical trials assessing EZH2i for the treatment of malignancies including ATRTs and MRTs are currently underway, with early signs of therapeutic benefits[22–24]. Therefore, understanding the mechanisms underlying the biological effects of EZH2i on SMARCB1-deficient tumors is crucial to developing therapeutic strategies that employ EZH2i.

More recently, the anti-tumor effects of EZH2i have been implicated in activating retroelements and viral mimicry, a cellular state characterized by loss of cancer cell fitness and the induction of innate and adaptive immune response, as well as interferon signaling[25–27]. However, the molecular mechanisms underlying how EZH2i initiates viral mimicry in ATRTs remains elusive. Here we profile the transcriptomic and epigenomic changes induced by EZH2i and identify that a major mechanism underlying the vulnerability of ATRT cells to EZH2i is the delayed activation of viral mimicry. Mechanistically, EZH2i induces the expression of genes enriched for intronic or 3′UTR IR-Alu elements, which triggers dsRNA formation and the MAVS-mediated dsRNA sensing pathway. Importantly, we identify a potential feedforward loop wherein many EZH2i-induced ISGs contain active IR-Alu elements, which may reinforce the viral mimicry response. Additionally, we demonstrate that EZH2i induces cytoplasmic ssDNA genome instability and micronuclei formation via LINE-1-mediated reverse transcription, thereby activating the cytosolic DNA sensing pathway. Thus, we found a unique vulnerability in genetically defined, SMARCB1-deficient tumors that is dependent on EZH2i-stimulated viral mimicry mediated by both cytosolic RNA and DNA sensing pathways.

## Results

### EZH2i induces delayed activation of interferon signaling in ATRT cells

To investigate the vulnerability of ATRT cells to EZH2i, we used UNC1999, an orally bioavailable chemical probe that inhibits the methyltransferase activity of EZH2[28]. A close chemical analog of UNC1999 is UNC2400, with the difference being that UNC2400 is unable to inhibit EZH2. Thus, UNC2400 was used as a negative control for any potential off-target effects of UNC1999 treatment[28]. As expected, UNC1999 treatment led to complete depletion of H3K27me3 levels without affecting the protein levels of EZH2 in four ATRT cell lines (BT12, BT16, CHLA02, and CHLA05) (Supplementary Fig. 1a). In contrast, UNC2400 treatment did not result in a significant reduction in either EZH2 or H3K27me3 levels. We next examined the growth potential of the four ATRT cell lines in response to UNC1999 treatment. We found that UNC1999 greatly inhibited the growth of all four ATRT cell lines in comparison to UNC2400 or DMSO, where DMSO was used as the vehicle control (Supplementary Fig. 1b). Next, we applied the CellTrace Violet (CTV) cell proliferation assay to examine the effects of UNC1999 on ATRT cell proliferation. Our analysis showed a minor decline in cell proliferation at day 3 post UNC1999 treatment in BT12 cells, while a greater reduction was observed at day 7 (Supplementary Fig. 1c). Furthermore, we found that the anti-proliferative effect of UNC1999 was dose-dependent and consistent across all ATRT cell lines tested. Our data together demonstrate that EZH2i leads to a time- and dose-dependent reduction in ATRT cell proliferation. To assess the impact of EZH2i on cell

viability in ATRT cells, we conducted Annexin V and propidium iodide (PI) labeling, followed by flow cytometry. Analysis of apoptosis in CHLA02 cells revealed a dose-dependent increase in apoptotic fractions (Annexin V-positive) following UNC1999 treatment, concomitant with a decrease in the percentage of viable cells (Annexin V and PI-negative) (Supplementary Fig. 1d–f). Moreover, this effect on cell death induction by UNC1999 was consistently observed across all tested ATRT cell lines (Supplementary Fig. 1e, f). In summary, our analyses of cell proliferation and apoptosis collectively suggest that impaired cell proliferation and increased cell death contribute to the growth inhibitory effects of EZH2 inhibition in ATRT cells.

To understand the mechanisms underlying the high sensitivity of ATRT cells to UNC1999 treatment, we performed RNA-sequencing (RNA-seq) on BT16 cells treated with UNC1999 or UNC2400 at day 4 and day 6. Differential expression analysis revealed that at both timepoints, UNC1999 treatment resulted in a greater number of upregulated genes than downregulated genes compared to UNC2400 treated controls (Fig. 1a). We performed pathway analysis on the upregulated genes following UNC1999 treatment at both timepoints. At day 4, we observed that pathways related to cell migration and differentiation were significantly enriched (Fig. 1b). Intriguingly, at day 6, the most enriched pathways were associated with cytokine response and interferon signaling, indicating the activation of a viral mimicry state[29] (Fig. 1c). We also found a higher number of upregulated genes at day 6 versus day 4 following UNC1999 treatment (Supplementary Fig. 2a). Notably, the top two significantly enriched pathways in UNC1999-treated cells at day 6 were the innate immune response and interferon signaling (Fig. 1d). These findings together suggest that interferon signaling pathway is activated as a delayed response following UNC1999 treatment. Based on the increased expression of their target genes, we predicted the transcription factors (TFs) involved in the upregulated genes induced by UNC1999. We observed an enrichment of the target genes of multiple interferon regulatory factors (IRFs) and STAT2, key TFs in the interferon signaling pathway, in cells treated with UNC1999 at day 6 (Fig. 1e and Supplementary Fig. 2b). These data also suggest a delayed and predominant activation of interferon signaling triggered by EZH2i.

To validate that EZH2i induces interferon signaling in ATRT cells, we examined the expression of interferon-stimulated genes (ISGs) via real-time qPCR following UNC1999 treatment. We found that in all four tested ATRT cell lines, the gene expression of four selected ISGs (*ISG15, DDX58, OASL,* and *OAS1*) was significantly upregulated in the UNC1999-treated samples compared to the UNC2400 controls (Fig. 1f). We then treated BT16 cells with lower doses of UNC1999 (1 μM or 2.5 μM) for 18 days and assessed ISG expression every 6 days. We observed a significant dose-dependent increase in ISG expression at day 6, with a more pronounced induction at day 12 and persisting until day 18 (Supplementary Fig. 2j). We also tested the effect of GSK343[30], another selective inhibitor of EZH2, on ISG activation. We observed that GSK343 induced ISG expression to similar levels compared to UNC1999 (Supplementary Fig. 2c). We also examined the effects of EZH2i on the expression of multiple interferon (IFN) genes. Our analysis revealed that treatment with either UNC1999 or GSK343 significantly upregulated the expression of the type I IFN gene IFNB1, as well as the type III IFN genes IFNL1 and IFNL3 (Supplementary Fig. 2d). We then extended our investigation to include two additional clinically relevant inhibitors: Valemetostat (DS-3201, an EZH1/2 inhibitor) and Tazemetostat (EPZ6438, an EZH2 inhibitor). Notably, both inhibitors demonstrated a significant induction of ISGs at both day 6 and day 12 following treatment (Supplementary Fig. 2e, f). Furthermore, we conducted EZH2 knockout experiments in BT16 and found that EZH2 deficiency induced significant growth inhibition and ISG activation (Supplementary Fig. 2g–i). Collectively, these data from both pharmacological and genetic inhibition approaches demonstrate that EZH2 inhibition triggers viral mimicry in ATRT cancer cells. Finally,

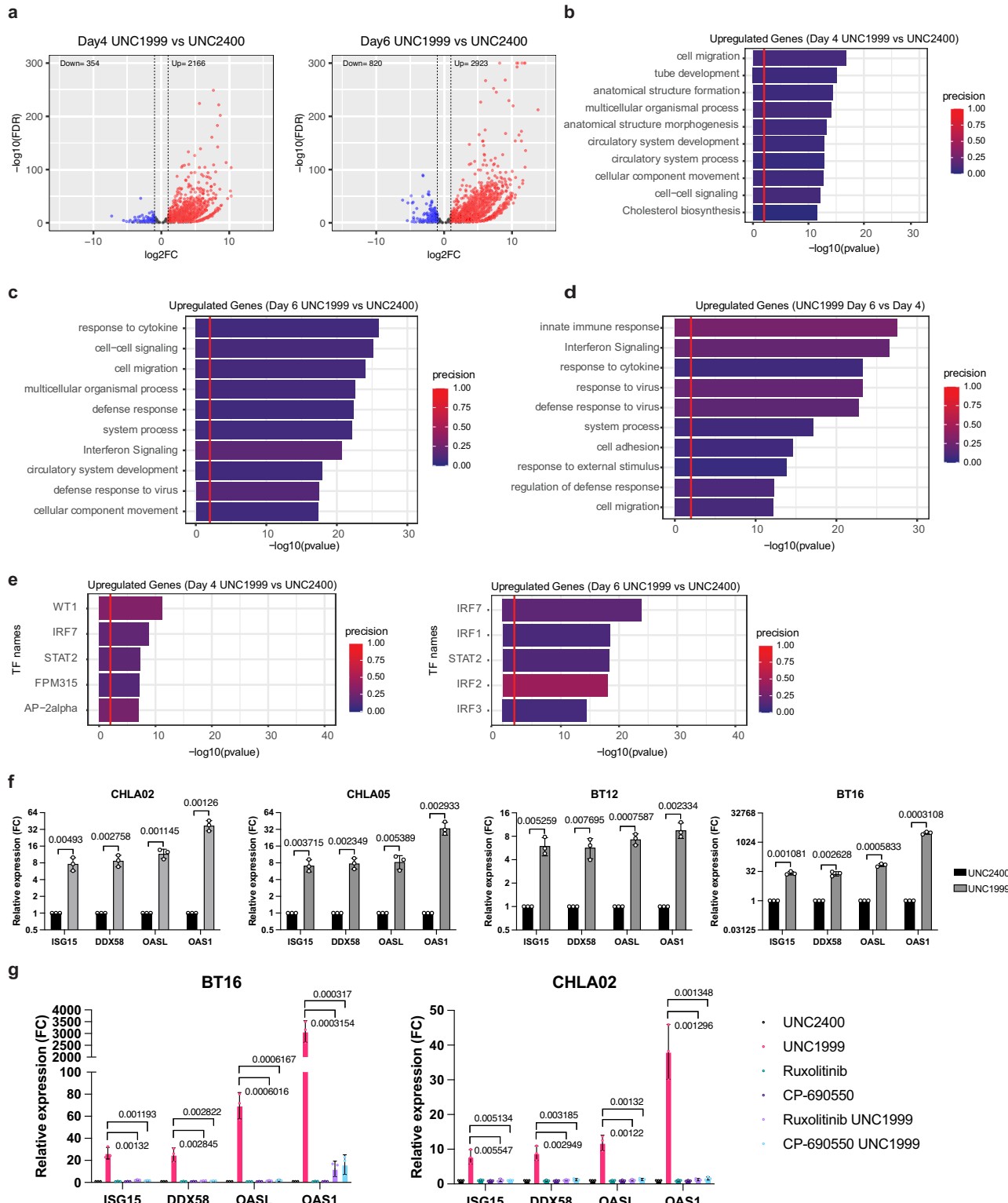

to examine whether the ISG activation is mediated through the JAK/STAT pathway, cells were treated with UNC1999 in the presence or absence of Ruxolitinib (JAK1/JAK2 inhibitor)[31] or CP-690550 (JAK1/JAK2/JAK3 inhibitor)[32]. Both Ruxolitinib and CP-690550 abolished the activation of ISGs in ATRT cells following UNC1999 treatment (Fig. 1g), demonstrating that the induction of ISGs is JAK-dependent. Together, our data illustrate that EZH2i induces delayed activation of interferon signaling in ATRT cells, and the expression of ISGs is mediated by the JAK/STAT pathway.

## EZH2i triggers RNA sensing pathway through derepressing IR-Alu elements

Next, we investigated whether EZH2i triggers interferon signaling by activating the dsRNA sensing pathway and inducing viral mimicry. The recognition of cytosolic dsRNA by pattern recognition receptors RIG-I/MDA5 initiates viral mimicry responses, including the upregulation of IFNs and downstream ISGs[33]. To test if UNC1999 induces interferon signaling through the dsRNA sensing pathway, we performed immunostaining with a monoclonal antibody against dsRNA to detect the

**Fig. 1 | UNC1999 treatment triggers delayed activation of interferon response.**
**a** Volcano plots showing the genes differential analysis statistics of UNC1999 ($n = 3$)
versus UNC2400 at day 4 (left) and at day 6 (right) in BT16 cell line. The $x$ axis
represents the $\log_2$FC of differential expression and the $y$ axis represents the
-$\log_{10}$(FDR) of DEGs. Blue dots represent downregulated genes and red dots
represent upregulated genes. Black dots represent genes that are not significantly
differentially expressed. Significance was determined by |$\log 2$FC| > 1 and FDR <
0.05. Negative binomial likelihood ratio test with BH (Benjamini–Hochberg)-cor-
rected for multiple testing. **b**–**d** Gene ontology (GO) analysis of significantly
upregulated genes in UNC1999 versus UNC2400 at day 4 (**b**), UNC1999 versus
UNC2400 at day 6 (**c**), and UNC1999 at day 6 versus day 4 (**d**). Precision represents
the overlap between the tested genes and the gene set of the terms. $P$-value is
calculated by the one-sided hypergeometric test followed by correction for

multiple testing. **e** Analysis of transcription factor binding site enrichments for
upregulated genes in UNC1999 versus UNC2400 at day 4 (left) and day 6 (right). $P$-
value is calculated by the hypergeometric test followed by correction for multiple
testing. **f** The expression of selected interferon-responsive genes in four indicated
ATRT cell lines with either UNC2400 or UNC1999 treatment was measured by
quantitative real-time PCR at day 6. **g** The CHLA02 (left) or BT16 (right) cell line was
treated with UNC1999 in the presence or absence of the JAK1/JAK2 inhibitor Rux-
olitinib (1 μM) or the JAK3 inhibitor CP-690550 (1 μM). The expression of four
indicated interferon-responsive genes was measured by quantitative real-time PCR
at day 6. Data are mean ± SD of three biologically independent replicates; $P$-value is
calculated by multiple unpaired $t$ tests (two-tailed) followed by correction for
multiple testing (**f**, **g**). Source data are provided as a Source Data file.

expression of dsRNA[34]. Our analysis revealed a significant increase in
dsRNA in ATRT cells following UNC1999 treatment (Fig. 2a, b). Upon
dsRNA expression, the RIG-I/MDA5 receptors detect its presence and
bind to the mitochondrial protein MAVS. This binding triggers signal
transduction that subsequently activates downstream IRFs, resulting
in the induction of IFN expression[33]. Importantly, the functional acti-
vation of MAVS is dependent on the formation of large MAVS prion-
like aggregates[35]. To investigate whether UNC1999 treatment activates
interferon signaling through MAVS, we isolated the mitochondrial
fraction of ATRT cells treated with UNC1999 or UNC2400. Then, we
employed semi-denaturing detergent agarose gel electrophoresis
(SDD-AGE) to detect MAVS aggregates. We identified a smear of SDS-
resistant high molecular weight MAVS aggregates in BT16 and CHLA02
cells treated with UNC1999, but not in the UNC2400 controls (Fig. 2c).
This indicates the functional activation of MAVS by UNC1999 treat-
ment. To further investigate whether the MAVS-mediated RNA sensing
pathway activates interferon signaling, we inactivated MAVS in BT16
cells via CRISPR-Cas9. MAVS-deficient cells exhibited a significant
reduction in ISG induction upon UNC1999 treatment compared to the
MAVS-wildtype cell line (Fig. 2d). These results suggest that the viral
mimicry induced by EZH2i is partially dependent on the MAVS-
mediated RNA sensing pathway.

Next, we sought to identify the retroelements that could be the
sources of immunogenic dsRNA following EZH2i. Differential analysis of
repeat expression from RNA-seq data revealed that a large number of
retroelements gained expression at day 4 and day 6 following UNC1999
treatment (Supplementary Fig. 3a). We further examined whether any
repeat classes were over-represented among the upregulated repeats
and found that SINE elements were enriched as the most significantly
upregulated repeats at both timepoints (Fig. 2e). In addition, we
examined the changes in the expression of major repeat families fol-
lowing UNC1999 treatment. Alu retroelements, the largest repeat family
among the upregulated repeats, were significantly enriched at both day
4 and day 6 following UNC1999 treatment (Supplementary Fig. 3b, c).
Our findings are consistent with a prior study identifying IR-Alu ele-
ments as the primary source of drug-induced immunogenic dsRNA[36].

IR-Alu elements can form dsRNA stem-loops via intramolecular
pairing[36]. Therefore, we examined the expression dynamics of all
annotated IR-Alu elements upon EZH2i. At both day 4 and day 6, a
higher number of IR-Alu repeats exhibited increased expression levels
following UNC1999 treatment, as compared to the UNC2400 control
(Fig. 2f). We then examined the changes in expression of each of the
two repeats within the IR-Alu pairs, as active transcription of both
repeats is required for dsRNA formation. Following the UNC1999
treatment, 532 and 1094 IR pairs exhibited increased expression in
both repeats at day 4 and day 6, respectively (Fig. 2g). We identified
these IR-Alu pairs upregulated in both repeats as UNC1999-induced
IRs, which can potentially make dsRNA. Notably, some of these IR-Alu
pairs have been experimentally validated as immunogenic given their
capacity to produce dsRNA that activates MDA5[36] (Fig. 2g). Moreover,
these experimentally validated IRs were statistically overrepresented

in the upregulated annotated IR pairs following UNC1999 treatment
(Fig. 2g). We immunoprecipitated dsRNA from total RNA in BT16 cells
using the J2 antibody and examined the enrichment of Alu elements in
dsRNA. qRT-PCR analysis targeting Alu consensus sequences revealed
significantly higher Alu enrichment in the dsRNA of UNC1999-treated
cells (Supplementary Fig. 3d). Additionally, using primers designed for
experimentally validated immunogenic IR-Alus upregulated upon
EZH2 inhibition, we observed higher enrichment in dsRNA from
UNC1999-treated cells compared to UNC2400 (Fig. 2h). Similar results
were obtained with primers targeting representative upregulated
annotated IR-Alu pairs (Fig. 2i). These findings collectively suggest that
IR-Alus induced by EZH2 inhibition can form dsRNA, potentially acti-
vating the dsRNA sensing pathway.

To investigate whether EZH2 directly targets the UNC1999-
induced IR pairs, we conducted genome-wide CUT&RUN (cleavage
under targets and release using nuclease) analysis[37]. As expected, we
observed a global depletion of the H3K27me3 signal in cells treated
with UNC1999 at both day 4 and day 6 (Supplementary Fig. 3e). Spe-
cifically, our analysis showed a robust deposition of H3K27me3 at
upstream regions of these IRs in the UNC2400 control, which was
depleted upon UNC1999 treatment (Supplementary Fig. 3f). In addi-
tion, the H3K27me3 depletion was accompanied by an increase in the
transcription levels of both the IR pairs and the regions surrounding
IRs (Supplementary Fig. 3g). Together, these findings suggest that
EZH2 directly targets the UNC1999-induced IR pairs.

## EZH2i upregulates intronic transcription that expresses IR-Alu elements

Next, we investigated the mechanism by which UNC1999 induces the
expression of IR-Alu elements. The majority of UNC1999-induced IR-
Alu elements originated from introns (65.8% at day 4 and 72.9% at day
6), followed by intergenic regions (18.3% at day 4 and 15.7% at day 6),
and 3′ UTR (13.3% at day 4 and 8.57% at day 6) (Fig. 3a). We observed a
significant enrichment of intronic and 3′ UTR IR-Alu elements, while
intergenic IR-Alu repeats were under-represented in the upregulated
IR-Alu repeats at both day 4 and day 6 following UNC1999 treatment,
as compared to the expected genomic distribution (Fig. 3a). This can
be explained by the H3K27me3 distribution analysis, indicating a larger
fraction of baseline H3K27me3 peaks overlapping with intronic
regions in the UNC2400-treated conditions (Supplementary Fig. 3h).
Out of the 617 UNC1999-induced IR-Alu elements at day 4, 298 (48%) of
them continued to be upregulated at day 6 (Supplementary Fig. 4a). In
addition, we found that 962 (76%) annotated IR-Alu elements were
induced only at day 6 following UNC1999 treatment. We further
examined the transcription direction (i.e., sense vs. anti-sense) of
UNC1999-induced intronic and 3′ UTR IRs relative to their associated
genes. Our analysis revealed that most IRs were transcribed in sense
with their overlapping genes (Fig. 3b and Supplementary Fig. 4b).
These findings suggest a strong correlation between the transcription
of UNC1999-induced IR-Alu elements and the activation of their
associated genes.

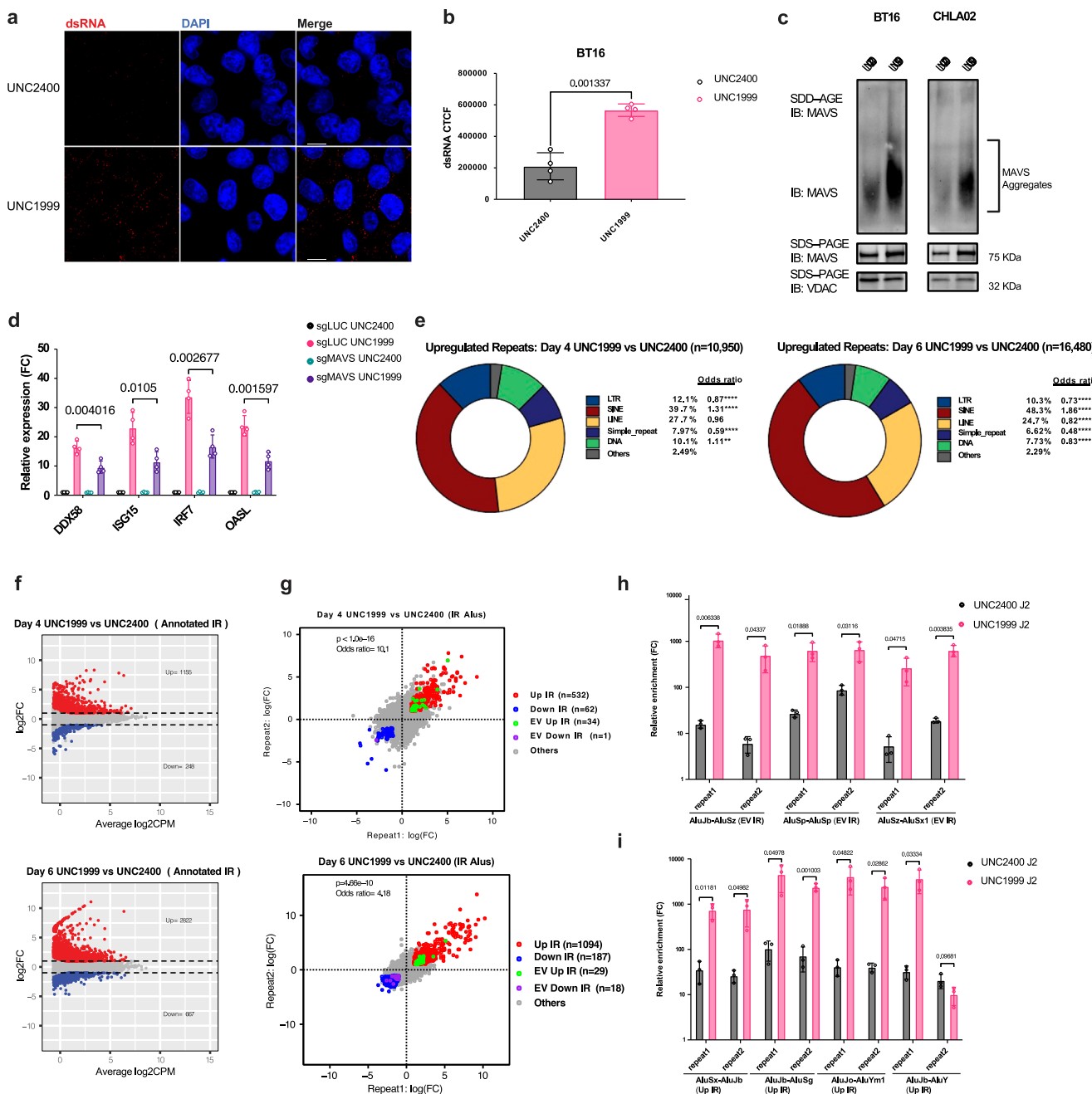

We hypothesized that EZH2i triggers excessive transcription of the IR-Alu-associated genes, which ultimately leads to the intronic transcription of IR-Alu elements. To test this hypothesis, we examined the expression dynamics and H3K27me3 signals of genes overlapping with upregulated intronic IRs, as well as the changes in the expression of introns within these genes following UNC1999 treatment. H3K27me3 depletion was detected at the promoters and gene bodies of the genes overlapping with upregulated intronic IRs following UNC1999 treatment, accompanied by an increase in transcription levels at both day 4 and day 6 (Fig. 3c and Supplementary Fig. 4c). In addition, we identified a negative correlation between changes in H3K27me3 signal and alterations in both gene expression and H3K4me3 signal at promoters of genes containing upregulated IR-Alu pairs (Supplementary Fig. 4d). Additionally, we observed a positive association between the dynamics of H3K4me3 signal at promoters and the expression change in these genes (Supplementary Fig. 4d). These data demonstrate that EZH2i induces upregulation of the genes

associated with UNC1999-induced intronic IRs. Notably, a significant increase in RNA-seq signal was found in the introns of genes containing the upregulated intronic IRs (Fig. 3d). Moreover, we observed a strong positive correlation in the expression changes between genes and their overlapping intragenic IR pairs following UNC1999 treatment (Fig. 3e). Together, these results indicate that EZH2i depletes H3K27me3 deposition at the promoters and gene bodies of IR-Alu associated genes, resulting in the accumulation of H3K4me3 at promoters and the transcriptional upregulation of these genes, as well as their intronic or 3′ UTR IR-Alu elements (Fig. 3h).

We further characterized the genes associated with UNC1999-induced intronic and 3′ UTR IRs. Analysis of these genes indicated that 62 were shared between the two timepoints, 46 were exclusively observed at day 4, and 145 were solely detected at day 6 (Supplementary Fig. 4a). Notably, we found that many genes associated with induced IRs were upregulated at both day 4 and day 6 following UNC1999 treatment (Supplementary Fig. 4e). The upregulated

**Fig. 2 | UNC1999 treatment induces RNA sensing pathway through activating Alu IRs. a** Confocal microscopy of anti-dsRNA (K1) immunofluorescence in BT16 cells treated with either UNC2400 or UNC1999. Cellular dsRNA was stained in red, and nuclei were stained in blue (DAPI). Scale bars, 10 µm. **b** Quantification of dsRNA performed by measuring corrected total cell fluorescence (CTCF), using ImageJ. Data are mean ± SD of $n = 4$ biologically independent replicates; unpaired t test with Welch's correction (two-tailed). **c** Representative immunoblot of MAVS aggregation assays analyzed by SDD-AGE in BT16 (left) and CHLA02 (right) cell lines with either UNC2400 or UNC1999 treatment. MAVS protein level was analyzed by SDS–PAGE. VDAC was used as a loading control in SDS–PAGE. **d** The BT16 cell line with or without sgRNA against MAVS was treated with either UNC2400 or UNC1999. The expression of four indicated ISGs was measured by qRT-PCR at day 6. Data are mean ± SD of $n = 4$ biologically independent replicates; multiple unpaired t tests (two-tailed) followed by correction for multiple testing. **e** Donut plots showing the proportions of repeat classes that are upregulated in RNA-seq UNC1999 versus UNC2400 at day 4 (top) and day 6 (bottom). Counts of repeat classes were compared with the whole-genome counts of repeat classes using the two-sided Fisher exact test to calculate the *p*-value and odds ratio. **$p < 0.05$; ****$p < 0.0001$. The p-values for day 4 plot (left) are 8.647e-07 for LTR, 0.51 for LINE, <2.2e-16 for SINE and Simple repeat, and 1.02e-3 for DNA. The p-values for the day 6 plot (right) are <2.2e-16 for LTR, SINE, LINE, and Simple repeat; and 7.6e-11 for DNA. **f** Mean average (MA) plots showing the upregulated Annotated IR-Alu elements in RNA-seq UNC1999 versus UNC2400 at day 4 (top) and day 6 (bottom). Red and

blue dots represent the upregulated and downregulated IR-Alu elements respectively. The $x$ axis represents the $\log_2$ count per million (CPM) in expression, and the $y$ axis represents the $\log_2$ Fold-change. Dots in gray color represents IR-Alu elements that are not significantly regulated. Significance was determined by $|log_2FC|$ >1 and FDR < 0.05. Negative binomial likelihood ratio test with BH (Benjamini–Hochberg)-corrected for multiple testing. **g** Inverted repeat Alu elements analysis depicted by scatter plots of the $\log_2$FC of the IR pairs (i.e., repeat1 and repeat2) of UNC1999 to UNC2400 at day 4 (top) and day 6 (bottom). Gray dots represent Annotated IR pairs that are not upregulated. Red and blue dots represent Annotated IR pairs that are upregulated (Up IR) and downregulated (Down IR) in RNA-seq respectively, and green and purple dots represent Experimentally validated (EV) IR-Alu elements that are upregulated (EV Up IR) and downregulated (EV Down IR) in RNA-seq respectively. Significance was determined by abs(log$_2$FC)>1 and FDR < 0.05 for both Alu elements. The count of the upregulated EV IR pairs was compared with the count of upregulated Annotated IR pairs using the two-sided Fisher exact test to calculate the odds ratio and p-value. **h, i** Error bar plots showing the enrichment of indicated transcripts of upregulated EV IR-Alus (**h**) or annotated IR-Alus (**i**) in the dsRNA species immunoprecipitated with J2 antibody from total RNA harvested from UNC1999- or UNC2400-treated BT16 cells. qRT-PCR was employed for analysis, with normalization to the corresponding input RNA). Data are mean ± SD of $n = 3$ biologically independent replicates; multiple unpaired t tests (two-tailed) followed by correction for multiple testing (**h, i**). Source data are provided as a Source Data file.

differentially expressed genes (DEGs) with induced IRs were also overrepresented among the total upregulated DEGs at day 4 and day 6 (Supplementary Fig. 4e). Pathway analysis of the upregulated genes associated with UNC1999-induced IRs uncovered that at day 4, the most enriched pathways were involved in metabolism or biosynthesis processes (Supplementary Fig. 4f). In contrast, at day 6, the most enriched pathways included viral defense response, interferon signaling, and innate immune response (Fig. 3f and Supplementary Fig. 4g). Additionally, intersecting genes co-induced with IR-Alus at day 4 and day 6 with the combined set of ISGs and IFN pathway genes revealed a notable and statistically significant overrepresentation at day 6 (Supplementary Fig. 4h), providing additional evidence that the upregulated genes associated with UNC1999-induced IR-Alus are enriched for ISGs. These findings suggest that certain ISGs associated with viral mimicry contain IR-Alu elements in their introns or 3' UTRs. These intragenic IR-Alu elements can themselves be a source of dsRNA, which in turn reinforces the viral mimicry response, thus forming a positive feedback loop (Fig. 3g, Supplementary Fig. 4i, j, k). Multiple genes, including DDX58, HERC6, and GBP4, were found to have active intronic or 3' UTR IR-Alu elements and exhibited increased transcription and chromatin remodeling, including depletion of H3K27me3 and accumulation of H3K4me3 at promoters (Fig. 3g and Supplementary Fig. 4i, j). For instance, *DDX58*, an ISG with multiple IR-Alu elements located in its introns, showed increased gene expression and IR-Alu expression at day 6 following UNC1999 treatment (Fig. 3g), which can potentially reinforce dsRNA expression and enhance dsRNA-induced viral mimicry.

Next, we extended our investigation to include an RNA-seq dataset comprising primary brain tumors, covering both SMARCB1-wildtype brain tumors and SMARCB1-deficient ATRTs. Despite a low tumor mutational burden, rhabdoid tumors including ATRTs demonstrate significant immune infiltration[38]. The heightened immunogenicity observed in rhabdoid tumors implies a potential elevation in the baseline expression of retroelements, which may prime these tumors for viral mimicry induction. Indeed, differential expression analysis revealed that ATRTs exhibit higher expression of retroelements, particularly Alu elements, compared to SMARCB1-wildtype brain tumors (Supplementary Fig. 5a, b). Further scrutiny of annotated IR-Alu elements known to form dsRNA[36], revealed a significantly greater number of these repeats with increased expression in ATRTs (Supplementary Fig. 5c, d). A significant proportion of these repeats exhibited upregulation on both strands (Supplementary Fig. 5e), suggesting potential

dsRNA formation. Additionally, the total expression of IR-Alu repeats was computed, demonstrating an overall increased expression in ATRTs compared to SMARCB1-wildtype brain tumors (Supplementary Fig. 5f). To strengthen our findings, we re-analyzed published RNA-seq data in a rhabdoid tumors cell line (I2A) with inducible SMARCB1 re-expression[38]. Analysis of annotated IR-Alu repeats revealed increased transcription of these elements in the SMARCB1-deficient condition (Supplementary Fig. 5g–j). These findings suggest a baseline activation of viral mimicry by expressed IR-Alu pairs in the SMARCB1-deficient condition. Notably, the ISG score, computed using the mean expression levels of the 38 ISGs defined by Liu et al.[39], showed a significant increase in ISG expression in the SMARCB1-deficient condition (Supplementary Fig. 5k). Collectively, our findings from both the primary brain tumor cohort and the I2A cell line suggest that the SMARCB1 deficiency may prime ATRTs for the induction of viral mimicry, a response that can be further enhanced upon EZH2 inhibition.

### EZH2i activates the cGAS-STING DNA sensing pathway
The partial rescue of ISG induction in MAVS-deficient cells (Fig. 2d) suggests that the activation of the interferon signaling is not solely mediated by the MAVS-mediated RNA sensing pathway. Other pathways, such as the cytosolic DNA-sensing cGAS-STING pathway, may also contribute to the induction of viral mimicry and interferon signaling[40]. Previous studies have demonstrated the DNA sensing pathway as a mechanistic link between DNA damage and immune responses[41–43]. Since EZH2i has been associated with increased genome instability[44–48], we hypothesized that UNC1999 could induce DNA damage and trigger the DNA sensing pathway.

To examine if EZH2i induces DNA damage in ATRT cells, we measured the nuclear foci of γH2A.X, a biomarker of cellular response to DNA double-strand breaks (DSB). The number of γH2A.X foci per cell was found to be higher in cells treated with UNC1999 compared to the UNC2400 control (Fig. 4a, b). In addition, we performed western blot and found that UNC1999 treatment increased the protein level of γH2A.X (Fig. 4e). These results suggest that EZH2i triggers DNA damage in ATRT cells. We further assessed micronuclei using confocal microscopy and found that their formation was elevated in cells treated with UNC1999 (Fig. 4a, c). This suggests that EZH2i leads to the increased micronuclei formation, which could arise from DNA damage[42]. Micronuclei are susceptible to nuclear envelope collapse, which exposes genomic dsDNA to cytosolic DNA sensor cGAS, initiating a cGAS–STING-dependent immune response through the

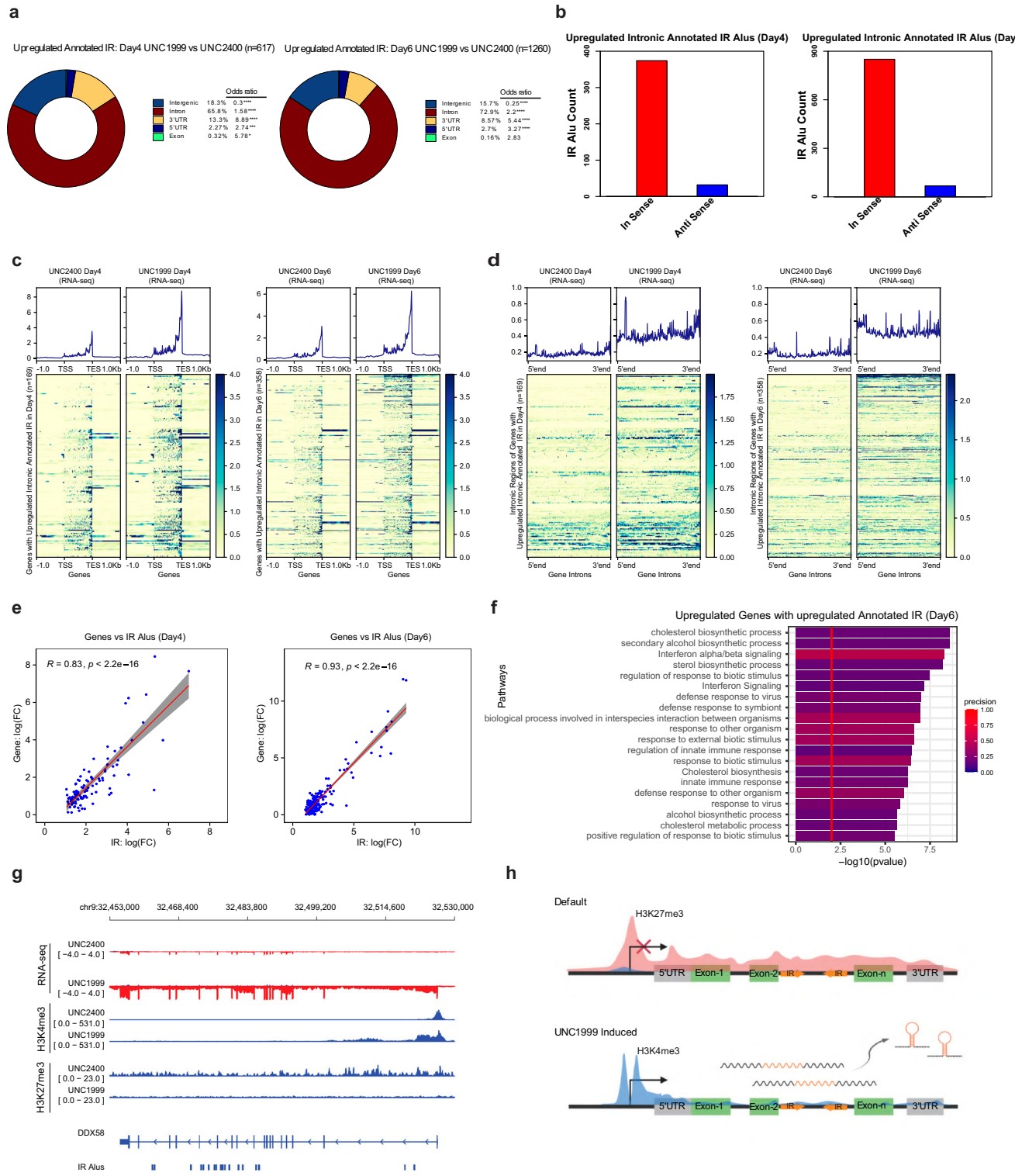

**Nature Communications** | (2024)15:9321

production of the second messenger cGAMP[49]. In line with this, in UNC1999-treated cells, micronuclei with γH2A.X staining were often cGAS-positive (Fig. 4a). Furthermore, we observed a higher fraction of cGAS-positive micronuclei formation in UNC1999-treated cells (Fig. 4d). We also found that cGAS-positive micronuclei were often co-localized with cytoplasmic dsDNA staining (Supplementary Fig. 6a), which is consistent with previous studies[42,43]. The protein levels of cGAS, IRF7, and phosphorylated-STAT1 (pSTAT1) increased in UNC1999-treated cells (Fig. 4e), suggesting that EZH2i activates the cGAS–STING-dependent DNA sensing pathway. To further assess the

dependency of viral mimicry induction on cGAS–STING pathway, we treated BT16 cells with UNC1999 in the presence or absence of H-151, a STING inhibitor[50]. We first established the efficacy of H-151 in BT16 cells by confirming the suppression of phosphorylated-STING (p-STING) and downstream effectors upon cGAMP transfection (Supplementary Fig. 6b). We then observed that the presence of H-151 significantly rescued the effect of UNC1999 on ISG induction (Fig. 4f), suggesting that STING activation is crucial to ISG induction. To further validate this, we knocked out cGAS via CRISPR-Cas9 and found that it effec-tively rescued the effect of UNC1999 on ISG induction (Fig. 4g and

**Fig. 3 | UNC1999 treatment induces intronic transcription that expresses IR-Alu elements. a** Donut plots showing the genomic distribution of the upregulated Annotated IR-Alu elements in RNA-seq UNC1999 versus UNC2400 day 4 (left) and day 6 (right). Counts of the upregulated Alu elements at genomic regions were compared with counts of all Annotated IR-Alu elements using the two-sided Fisher exact test to calculate the p-value and odds ratio. $*p < 0.05$; $****p < 0.0001$. The p-values for day 4 plot (left) are <2.2e-16 for Intergenic, 4.187e-08 for Intron, <2.2e-16 for 3′ UTR, 9.276e-04 for 5′ UTR, and 4.816e-02 for Exon. The p-values for the day 6 plot (right) are <2.2e-16 for Intergenic, Intron, and 3′ UTR; 7.147e-09 for 5′ UTR; and 0.159 for Exon. **b** Counts of the upregulated intronic Annotated IR-Alu elements at day 4 (left, $n = 406$ Alu elements) and day 6 (right, $n = 918$ Alu elements) that have transcription in sense and anti-sense of their overlapping genes. **c** DeepTool's aggregate profile plot (top) and heatmap (bottom) showing the RNA-seq signal in UNC1999 and UNC2400 samples of gene body regions as well as ±1 Kb up/downstream of the TSS and TES of the genes overlapping with upregulated intronic IRs in UNC1999 versus UNC2400 in day 4 (left, $n = 169$) and day 6 (right, $n = 358$). The orientation and strand of the RNA-seq signal are based on gene transcriptional orientation. **d**, DeepTool's aggregate profile plot (top) and heatmap (bottom) of the RNA-seq signal in UNC1999 and UNC2400 samples of intronic regions of genes that overlap with upregulated intronic IRs in UNC1999 versus UNC2400 at day 4 (left, $n = 169$) and day 6 (right, $n = 358$). 5′end and 3′end labels are the 5′ end of the first intron and the 3′ end of the last intron in the gene respectively. The orientation and strand of the RNA-seq signal are based on gene transcriptional orientation. **e**, Scatter plots showing the Pearson correlation in the $\log_2$-fold change between genes and their overlapping upregulated Intragenic (Intronic and 3′UTR) IR pairs in UNC1999 versus UNC2400 at day 4 (left, $n = 112$) and day 6 (right, $n = 214$). For genes with multiple IR pairs, the IR pair with the highest FC was selected for this analysis. R represents the Pearson correlation coefficient, and the Pearson correlation p-value is calculated using the two-sided $t$ test. **f** Pathway analysis of upregulated genes ($n = 84$) that overlap with upregulated intragenic IRs in UNC1999 versus UNC2400 in day 6. P-value is calculated using the one-sided hypergeometric test followed by correction for multiple testing. **g** Genome track plot showing signal of RNA-seq and H3K4me3 and H3K27me3 CUT&RUN marks at the DDX58 locus at day 6. DDX58 is significantly upregulated at day 6 UNC1999 versus UNC2400. The plot also includes at the track for the upregulated IR-Alu elements overlapping with the DDX58 gene. RNA-seq signal was plotted from the two strands separately. The RNA-seq forward strand signal was plotted in blue and positive range, while the RNA-seq reverse strand signal was plotted in red and negative range. RNA-seq tracks are plotted in log scale. The log scale of the forward strand signal was calculated as $log(signaltrack+1)$, while the log scale of the reverse strand signal was calculated as $-log(signaltrack+1)$. **h** Schematic representation of H3K27me3 deposition of IR associated genes by default and active transcription of intronic IR-Alu pairs upon UNC1999 treatment. Created in BioRender. De Carvalho, D. (2022) BioRender.com/e90j436.

Supplementary Fig. 6c). Notably, the double knockout of cGAS and MAVS abolished the ISG induction by UNC1999 treatment (Fig. 4g). Thus, the viral mimicry response activated by EZH2i is dependent on both the MAVS-mediated RNA sensing pathway and the cGAS-mediated DNA sensing pathway (Fig. 4h).

## EZH2i induces genome instability through L1-mediated reverse transcription

Next, we sought to investigate the mechanism underlying the genome instability induced by EZH2i. Retrotransposition mediated by long interspersed element-1 (LINE-1 or L1) and Alu elements has been implicated in the generation of insertional mutations and post-insertion-based structural rearrangements[51–54]. The ORF2 of L1 retrotransposons encodes enzymatic machinery including endonuclease and reverse transcriptase, which mediates the reverse transcriptionof both autonomous and non-autonomous elements[55]. Our RNA-seq analysis of repeat expression revealed that 62 full-length L1s (at least 6 kb in length) with increased transcription were significantly enriched among the total upregulated L1s at day 4 post UNC1999 treatment (Supplementary Fig. 7a, b). Of these, 42 L1s remained upregulated at day 6. These upregulated full-length L1s, annotated as the L1p family, showed a broad increase in transcription level from the 5′UTR to the 3′UTR (Supplementary Fig. 7c, d). In addition, we observed that the increased expression of L1s was accompanied by an increase in H3K4me3 and a depletion of H3K27me3 (Supplementary Fig. 7d, e). Specifically, a subset of upregulated full-length L1s exhibited H3K4me3 enrichment at their promoter regions, indicating active transcription following UNC1999 treatment (Supplementary Fig. 7f). Furthermore, both western blotting and immunostaining revealed an upregulation in the expression of L1 ORF1p under UNC1999-treated conditions (Supplementary Fig. 7g, h). Notably, UNC1999 treatment led to a depletion of H3K27me3 staining in nearly all cells, with the majority displaying increased expression of both STING and L1 ORF1p (Fig. 5a, b). These results collectively confirm that a specific subset of derepressed full-length L1s, triggered by EZH2 inhibition, undergo active expression. This subset may potentially serve as a crucial source of enzymatic machinery for L1-mediated reverse transcription and activate the DNA sensing pathway.

To investigate further, we examined the levels of single-stranded DNA (ssDNA) using an ssDNA-specific antibody following UNC1999 treatment. We found that UNC1999-treated cells exhibited a significantly higher number of ssDNA puncta per cell than those treated with UNC2400 (Fig. 5d). To investigate whether the ssDNA contain L1 elements, we extracted ssDNA species as previously desbribed[56], revealing a significant enrichment of L1 sequences in the UNC1999-induced ssDNA (Fig. 5c). We further assessed the dependence of ssDNA induction on LINE-1 activity by generating LINE-1-deficient cells using sgRNAs targeting the 5′UTR or ORF1 of L1. A detailed bioinformatics analysis of the specific L1 loci targeted by gRNAs suggested that the gRNA sequences are conserved across multiple L1 families, including L1H, L1PA2, and L1PA3, with a higher specificity for L1H (Supplementary Fig. 8a, b). We then demonstrated a significant decrease in LINE1 transcription and ORF1p protein levels following CRISPR-mediated genome editing (Supplementary Fig. 8c–e). We found a significant attenuation in the induction of ssDNA species in the LINE-1-deficient cells upon EZH2 inhibition (Fig. 5d). Consequently, we found that the activation of ISGs following EZH2 inhibition was significantly abolished with the inactivation of the LINE-1 elements (Fig. 5e). We next examined whether increased levels of ssDNA arise due to L1-mediated reverse transcription through employing reverse-transcriptase (RT) inhibitors known to inhibit L1 reverse transcription[57,58]. The addition of RT inhibitors attenuated the increase of ssDNA levels induced by UNC1999 treatment (Fig. 6a, b), indicating the formation of cytoplasmic ssDNA is dependent on L1-mediated reverse transcription.

We then performed γH2A.X immunostaining and western blot to measure DNA damage in the presence or absence of RT inhibitors. We found that RT inhibitors significantly rescued the effect of UNC1999 on DNA damage (Fig. 6c, d, g). In addition, we observed that the higher fraction of micronuclei-positive cells and the cGAS-positive micronuclei cells induced by UNC1999 treatment were abolished with the addition of RT inhibitors (Fig. 6c, e, f). Finally, the presence of reverse-transcriptase inhibitors attenuated the activation of key components of the cGAS-mediated DNA sensing pathway including cGAS, IRF7, and pSTAT1 upon UNC1999 treatment (Fig. 6g). Consequently, we found that the induction of ISG by UNC1999 was significantly reduced with the addition of RT inhibitors (Fig. 6h). In conclusion, our findings reveal that EZH2i upregulates the expression of full-length LINE-1 elements, which induces ssDNA formation, genome instability, and cytosolic DNA sensing pathway in a process dependent on reverse transcriptase activity.

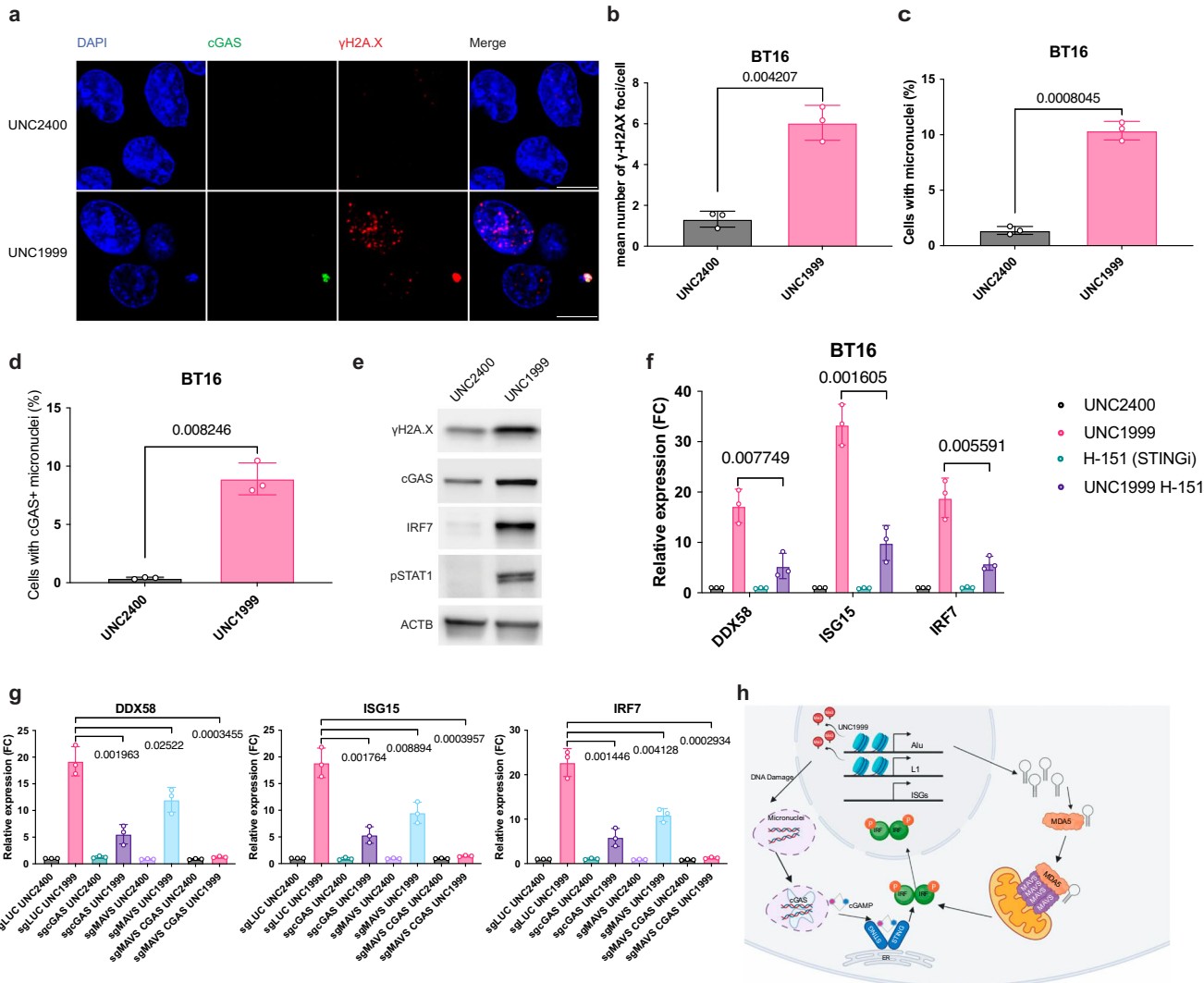

**Fig. 4 | UNC1999 treatment triggers cGAS-STING DNA sensing pathway.**
**a** Confocal microscopy of anti-cGAS and anti-γH2A.X immunofluorescence in BT16 cells treated with either UNC2400 or UNC1999. cGAS was stained in green, γH2A.X was stained in red, and nuclei were stained in blue (DAPI). Scale bars, 10 μm.
**b–d** Quantification of γH2A.X foci per cell (**b**), the percentage of cells with micronuclei (**c**), and the percentage of cells with cGAS-positive micronuclei (**d**) in **a**.
**e** Representative immunoblots showing the expression level of γH2A.X, cGAS, IRF7 and pSTAT1 with either UNC2400 or UNC1999 treatment in BT16 cell line. **f** The BT16 cell line was treated with UNC1999 in the presence or absence of the STING inhibitor H-151 (1 μM). The expression of indicated interferon-responsive genes was measured by quantitative real-time PCR at day 6. **g** The BT16 cell line with sgRNA

against LUC, MAVS, cGAS, or both cGAS and MAVS was treated with either UNC2400 or UNC1999. The expression of indicated interferon-responsive genes was measured by quantitative real-time PCR at day 6. **h** Schematic representation of viral mimicry triggered by UNC1999 treatment dependent on both MAVS-mediated RNA sensing pathway and cGAS-mediated DNA sensing pathway. Created in BioRender. De Carvalho, D. (2022) BioRender.com/t86f257 Data are mean ± SD of three biologically independent replicates (**b**, **c**, **f**, **g**); *P*-value is calculated by unpaired t test with Welch's correction (two-tailed) (**b**, **c**) or multiple unpaired *t* tests (two-tailed) followed by correction for multiple testing (**f**, **g**). Source data are provided as a Source Data file.

## Discussion

Pharmacological inhibitors targeting EZH2 in SMARCB1-deficient tumors have demonstrated promising outcomes in both preclinical and clinical studies[17,18,22–24]. Tazemetostat, a specific EZH2 inhibitor, has been granted accelerated approval by the US FDA for the treatment of SMARCB1-deficient cancers. Dissecting the mechanisms of action of EZH2i in SMARCB1-deficient tumors is crucial for optimizing treatment plans and achieving better outcomes. Our findings indicate that a major mechanism underlying the anti-tumor effects of EZH2i in ATRTs is the induction of viral mimicry response, which holds potential implications for the clinical activity of targeting EZH2 in ATRT patients. A recent report has shown high levels of T cell and myeloid cell infiltration in rhabdoid tumors, despite their low mutation burden[38]. Our results suggest that SMARCB1 deficiency may predispose these tumors

to viral mimicry, a state that can be enhanced with EZH2 inhibition. This could provide the scientific rationale for evaluating the combination of EZH2i and immunotherapy for the treatment of these tumors. Further studies are necessary to investigate the effects of EZH2 inhibitors on the tumor microenvironment and their ability to enhance immunotherapy including immune checkpoint blockade in SMARCB1-deficient tumors.

Our data suggest that intronic and 3' UTR IR-Alu elements are the major sources of dsRNA that stimulates viral mimicry triggered by EZH2i. Of note, transcriptomic and epigenomic analyses indicate that IR-Alu expression is driven by the excessive transcription of the genes associated with these elements, which become derepressed upon the removal of repressive H3K27me3 marks. These results contrast a previously discovered mechanism, showing that DNA

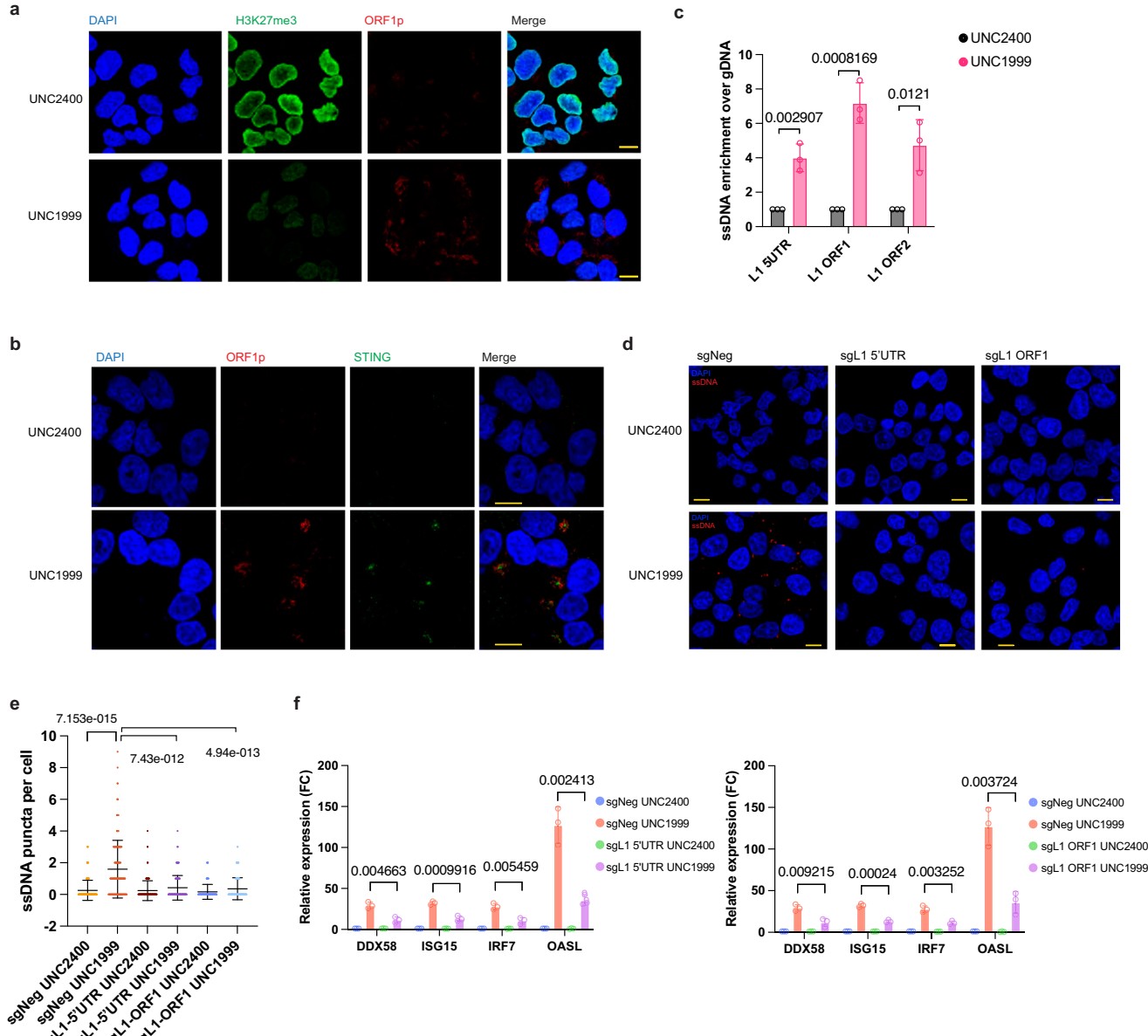

**Fig. 5 | The UNC1999-induced DNA sensing is dependent on LINE-1 activity.**
**a** Confocal microscopy of anti-H3K27me3 and anti-L1 ORF1p immunofluorescence in BT16 cells treated with either UNC2400 or UNC1999. H3K27me3 was stained in green, L1 ORF1p was stained in red, and nuclei were stained in blue (DAPI). Scale bars, 10 μm. **b** Confocal microscopy of anti-STING and anti-L1 ORF1p immunofluorescence in BT16 cells treated with either UNC2400 or UNC1999. STING was stained in green, L1 ORF1p was stained in red, and nuclei were stained in blue (DAPI). Scale bars, 10 μm. **c** Error bar plots showing the enrichment of indicated regions of LINE-1 elements in the ssDNA species extracted from UNC1999- or UNC2400-treated BT16 cells. qRT-PCR was employed for analysis, with normalization to the corresponding gDNA. **d** Confocal microscopy of anti-ssDNA

immunofluorescence in BT16 cell line with sgRNAs against negative control, L1-5′ UTR, or L1-ORF1 at day 4 post the UNC1999 or UNC2400 treatment. ssDNA was stained in red, and nuclei were stained in blue (DAPI). Scale bars, 10 μm.
**e** Quantification of ssDNA puncta per cell in **d**. with a sample size of 150 cells as a stopping point for all conditions. **f** The expression of four indicated interferon-responsive genes in BT16 cell line with sgRNAs against negative control, L1-5′UTR (left), or L1-ORF1 (right) was measured by qRT-PCR at day 6 post the UNC1999 or UNC2400 treatment. Data are mean ± SD of three biologically independent replicates (**c**, **f**) or 150 data points (**e**); *P*-value is calculated by multiple unpaired t tests (two-tailed) followed by correction for multiple testing (**c**, **f**) or unpaired *t* test with Welch's correction (two-tailed) (**e**). Source data are provided as a Source Data file.

demethylating agents triggers the cryptic transcription of a subset of intronic and intergenic IR-Alu elements[36]. These IR-Alu elements are usually located downstream of orphan CGIs and become derepressed upon DNA demethylation. These findings together indicate that DNA methylation and H3K27me3 preferentially target different IR-Alu elements in the human genome. The mutual exclusivity between DNA methylation and H3K27me3 in silencing IR-Alu elements could be explained by the antagonistic activity of DNA methylation in PRC2 recruitment. The interplay between DNA methylation and H3K27me3 could play important roles in the

regulation of retroelements and dsRNA formation. Whether the combination of DNMTi with EZH2i would achieve a broader activation of IR-Alu elements and increased expression of dsRNA is worth further investigations. Moreover, our analysis reveals that the intronic and 3′ UTR regions of ISGs activated by EZH2i contain IR-Alu elements that may produce dsRNA, leading to an enhancement of viral mimicry via a feedforward mechanism. This could have far-reaching implication in innate immunity, as ISGs seem to increase interferon response through the production of endogeneous dsRNA.

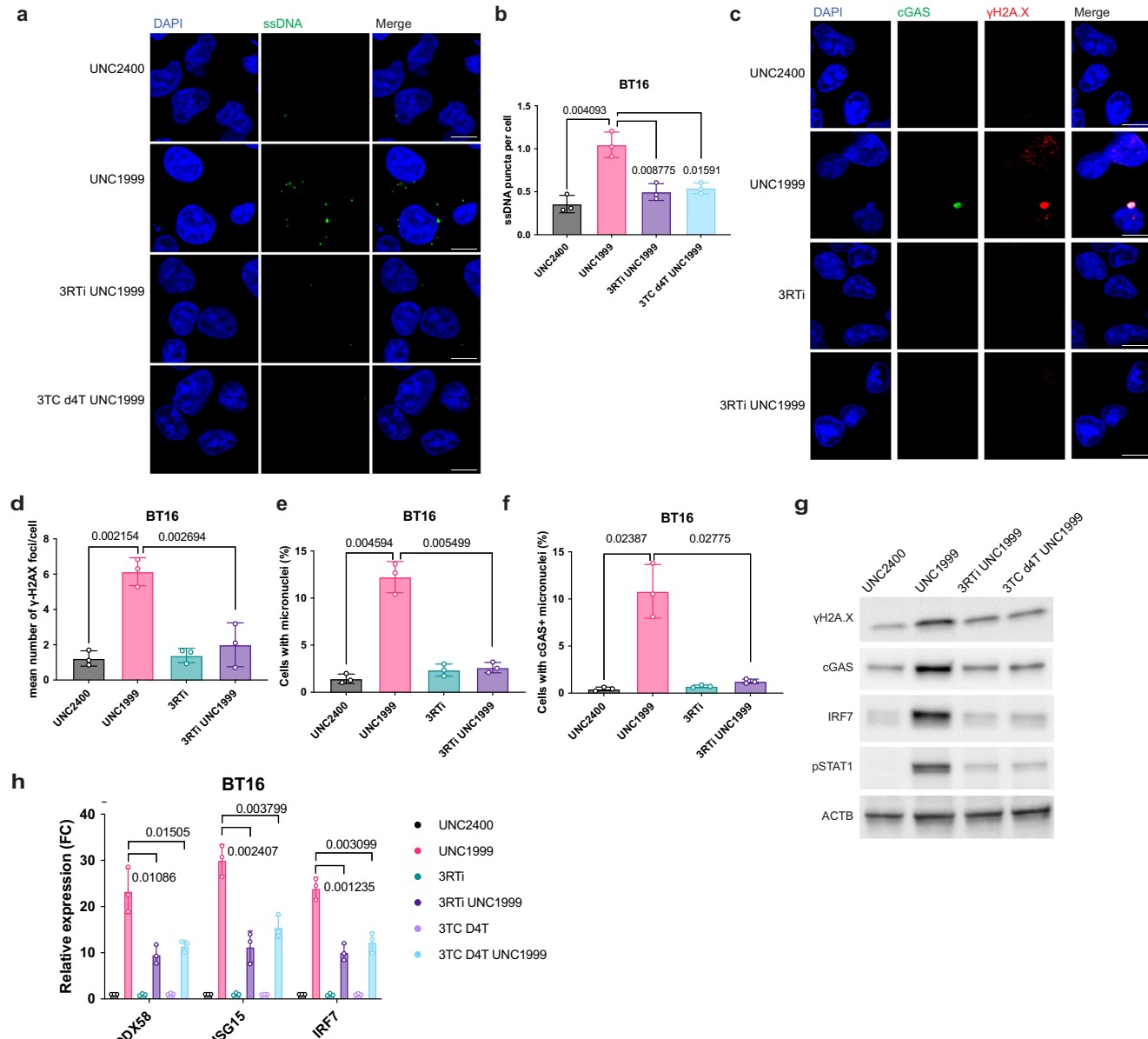

**Fig. 6 | UNC1999 treatment induces DNA damage through L1-mediated reverse transcription. a** Confocal microscopy of anti-ssDNA immunofluorescence in BT16 cells treated with UNC1999 in the presence or absence of RT inhibitors. ssDNA was stained in green, and nuclei were stained in blue (DAPI). 3RTi represents the combination of Zidovudine (AZT), Didanosine (ddI, DDI), Nevirapine (NVP). Scale bars, 10 μm. **b** Quantification of ssDNA puncta per cell in **a**. **c** Confocal microscopy of anti-γH2A.X immunofluorescence in BT16 cells treated with UNC1999 in the presence or absence of RT inhibitors. Scale bars, 10 μm. **d**–**f** Quantification of γH2A.X foci per cell (**d**), the percentage of cells with micronuclei (**e**), and the percentage of cells with cGAS-positive micronuclei (**f**) in **c**. **g** Representative immunoblots showing the expression level of γH2A.X, cGAS, IRF7, and pSTAT1 in BT16 cell line treated with UNC1999 in the presence or absence of RT inhibitors. **h** The BT16 cell line was treated with UNC1999 in the presence or absence of RT inhibitors. The expression of indicated interferon-responsive genes was measured by quantitative real-time PCR at day 6. Data are mean ± SD of three biologically independent replicates (**b**, **d**, **e**, **f**, **h**); *P*-value is calculated by unpaired t test with Welch's correction (two-tailed) (**b**, **d**, **e**, **f**) or multiple unpaired *t* tests (two-tailed) followed by correction for multiple testing (**h**). Source data are provided as a Source Data file.

Our research shows that EZH2i not only activates the dsRNA sensing pathway, but also triggers cytoplasmic ssDNA, genome instability and formation of micronuclei through L1-mediated reverse transcription, leading to the activation of the cGAS-STING pathway and viral mimicry state. In our study, we observed that inhibiting EZH2 leads to the induction of DSBs, as evidenced by increased γ-H2AX foci formation, which could be rescued with the addition of RT inhibitors. However, direct evidence of residual genomic scars from DSBs remains elusive, as whole-genome sequencing (WGS) struggles to detect genomic instability at the necessary resolution. Future studies should employ higher-resolution techniques, such as long-read

sequencing or single-cell sequencing, to identify potential DSBs and their remnants. These methods could complement our findings and offer a more comprehensive understanding of the impact of EZH2i on L1 activity and genome instability. In addition, our findings expand a previous report of EZH2i activating the dsRNA-STING-ISG response in prostate cancer models[26], providing a deeper understanding of viral mimicry induction by EZH2i. Indeed, co-depletion of MAVS and cGAS fully rescues ISG induction, indicating that viral mimicry is dependent on both dsRNA and DNA sensing pathways.

Our reverse transcriptase inhibition and L1 knockout/down experiments suggest that the presence of cytoplasmic ssDNA and

subsequent DNA sensing activation following EZH2i depend on L1 reverse transcriptase activity. Recent studies by Thawani et al.[59] and Baldwin et al.[60] provided comprehensive insights into the structural basis and molecular mechanism of the L1 retrotransposon. They demonstrated that the cytosolic reverse transcriptase activity of ORF2p can generate RNA:DNA hybrids, leading to the activation of DNA sensing pathway and downstream interferon signaling. These findings align with our discovery that cytoplasmic ssDNA induced by UNC1999 relies on the reverse transcriptase activity of L1 elements. However, our CUT&RUN and RNA-seq analyses did not pinpoint the specific L1 subfamily involved and might not have the resolution to detect autonomous L1Hs that could potentially be activated by EZH2i. The relatively ancient L1PA elements upregulated by EZH2i may lack the functional reverse transcriptase activity necessary for retro-transposition. It is conceivable that L1PA elements upregulated by EZH2i could exploit the reverse transcriptase produced by L1Hs expressed at basal levels to mobilize in a nonautonomous manner. These limitations should be considered when extrapolating our results to active L1 elements involved in viral mimicry.

Altogether, our study identifies the activation of viral mimicry as a fundamental mechanism underlying the vulnerability of ATRT cells to EZH2i. Specifically, IR-Alu elements from intragenic regions are characterized as the primary source of immunogenic retroelements, producing dsRNA through increased expression of their associated genes upon EZH2i. Moreover, our results suggest a positive feedback loop where the viral mimicry state is reinforced as certain ISGs activated by EZH2i contain IR-Alu sequences. Finally, our data show that EZH2i triggers genome instability through L1-mediated reverse transcription and activates the cGAS-STING pathway, working in conjunction with the dsRNA sensing pathway to drive viral mimicry induction.

# Methods

## Pharmacological treatment

The EZH2 inhibitors UNC1999 (Cayman Chemical 14621-50), GSK343 (Sigma-Aldrich SML0766), Valemetostat (DS-3201) (Selleck Chemicals S8926), and Tazemetostat (EPZ-6438) (Selleck Chemicals S7128) were reconstituted in DMSO as 10 mM stocks and stored at −80 °C. UNC2400 (gift from SGC), a negative control for UNC1999, was reconstituted in DMSO as 10 mM stock and stored at −80 °C. The JAK inhibitors Ruxolitinib (InvivoGen tlrl-rux) and CP-690550 (InvivoGen tlrl-cp69) were reconstituted in DMSO, both as 20 mM stocks and stored at −20 °C. The STING inhibitor H-151 (gift from SGC) was reconstituted in DMSO as 10 mM stocks and stored at −80 °C. The reverse-transcriptase (RT) inhibitors Stavudine (D4T) (Selleck Chemicals S1398), Lamivudine (3TC) (Selleck Chemicals S1706), Zidovudine (AZT) (Selleck Chemicals S2579), Didanosine (ddI, DDI) (Selleck Chemicals S1702), Nevirapine (NVP) (Selleck Chemicals S1742) were reconstituted in DMSO or water as 50 mM stocks according to the manufacturer's instructions and stored at −80 °C.

For in vitro experiments, cells were seeded at 50,000 cells/mL for 6–12 h, followed by UNC1999 (1 μM, 2.5 μM, or 5 μM) or UNC2400 (2.5 μM or 5 μM) treatment. For treatment with JAK inhibitors, Ruxolitinib or CP-690550 was added at a concentration of 1 μg/mL. For STING inhibition, H-151 was added at 1 μM. A combination of 1 μM D4T and 10 μM 3TC or a combination of 5 μM AZT, 5 μM ddI, and 5 μM NVP was used for treatment with RT inhibitors.

## Cell culture

ATRT cell line CHLA02-ATRT (ATCC, Cat# CRL-3020) and CHLA05-ATRT (ATCC, Cat# CRL-3037) were obtained from American Type Culture Collection (ATCC, Teddington, UK) and cultured in DMEM/F12 supplemented with 20 ng/mL FGF, 20 ng/mL EGF, and 1x B27 supplement. BT12 and BT16, which were kind gifts from Dr. Annie Huang at Hospital for Sick Children, were cultured in RPMI-1640 supplemented with 10% FBS.

## Cell count and cell proliferation assay analysis

Relative cell growth was measured by counting viable cells. Viable cell numbers harvested from all treatment conditions were counted using the automated cell-counting machine Vi-CELL XR (Beckman Coulter) following the manufacturer's instructions.

Cell proliferation was measured by CellTrace Violet (CTV) assay. Briefly, at day 0 prior to drug treatment, cells were stained with CTV dye (ThermoFisher Scientific C34557) according to the manufacturer's instructions. At day 3 and day 7 post UNC1999 or UNC2400 treatment, cells were harvested and washed with FACS washing buffer (PBS, 0.5% BSA, 2 mM EDTA). CTV fluorescence was measured by flow cytometry and analyzed using FlowJo software.

## Apoptosis assay

Cellular apoptosis was assessed through annexin V and propidium iodide (PI) staining using annexin V-FITC Apoptosis Detection Kit (BD Pharmingen 556547), according to the manufacturer's protocol. FITC and PI fluorescence was measured by flow cytometry and analyzed using FlowJo software.

## RNA-seq and data analysis

BT16 cells were treated with UNC1999 or UNC2400 at 5 μM and harvested at day 4 and day 6 for total RNA isolation. 12 total RNA samples from three biological replicates were extracted using the RNeasy Plus Mini Kit (QIAGEN, catalog No. 74104) and then quantified by qubit RNA kit (Life Technologies) and quality assessed by Agilent Bioananlyzer. All samples had an RNA integrity number greater than 9. Libraries were prepared using TruSeq Stranded Total RNA kit (Illumina). 200 ng of each RNA sample was ribosomal RNA depleted using Ribo-zero Gold rRNA beads. The RNA samples were fragmented following purification. The cleaved RNA fragments were copied into first strand cDNA using reverse transcriptase and random primers. This is followed by second strand cDNA synthesis using RNase H and DNA Polymerase I. A single 'A' nucleotide was added, followed by adapter ligation. The resulting product was then purified and amplified with PCR to create cDNA libraries. Final cDNA libraries were size validated using Agilent Bioanalyzer or Tapestation and concentration validated by qPCR (Kapa Biosystems/Roche). All libraries were normalized and pooled together, denatured with 0.2 N NaOH and diluted to a final concentration of 250 pM. Pooled libraries were loaded onto an Illumina Novaseq V1.5 cartridge for cluster generation and sequenced by the Princess Margaret Genomics Centre on an Illumina Novaseq 6000 instrument (Illumina). Paired-end 101 bp protocol was used to achieve around 60 million reads per sample.

Raw reads were trimmed using TrimGalore tool (http://www.bioinformatics.babraham.ac.uk/projects/trim_galore/) to remove adapters and low-quality bases. Bases with quality score less than 30 were removed, and trimmed reads that have length less than 40 bp were removed. Trimmed reads were aligned to the human genome reference hg38 using STAR tool v.2.5.2[61]. Aligned reads were sorted based on the genomic coordinates using the sort command from samtools package v.1.9[62]. Duplicate reads were marked using Picard tools v.1.9.1 (http://broadinstitute.github.io/picard/) with duplicate parameters marked as "lenient" and then removed using samtools. Genome track bigwig files were created for the forward and reverse strands using the bamCoverage command from the deepTools package[63].

RNA-seq differential analysis for genes was performed to assess the deferentially up/downregulated genes. The aligned reads were counted for genes using the featureCount command[64] from the sub-read tools v.1.6.2. The featureCount command was used with the following setting (-p -B --ignoreDup -O -s 2). Differential analysis for gene counts was performed using the edgeR library[65]. Significantly upregulated/downregulated genes were determined by abs(log$_2$FC)>1 and FDR < 0.05. Pathway analysis for the upregulated genes was performed using the R package gprofiler2[66,67].

RNA-seq differential analysis at repeat elements was also performed to assess the up/downregulated repeat elements. Aligned reads at repeat loci were counted for each sample using the feature-Count command from the subread tools. We used a gene transfer format (GTF) file for the whole genome repeat loci downloaded from the UCSC database. Reads were counted for each repeat locus from the two RNA-seq stands separately to assess the differential analysis of repeat loci by the two strands separately. The featureCount command was used with the following setting (-M -p -B --ignoreDup -f). Differential analysis for repeat loci counts was performed using the edgeR library. Significantly upregulated/downregulated repeat loci were determined by abs($\log_2$FC)>1 and FDR < 0.05.

Differential analysis for ATRT samples[21] ($n = 11$) versus SMARCB1-wildtype tumor samples ($n = 14$) was performed to evaluate the upregulated and downregulated repeats. Tumor total RNA were collected through the Rare Brain Tumor Consortium and Registry (www.rarebraintumorconsortium.ca) with informed consent as per protocols approved by the Research Ethics Board of the Hospital for Sick Children and sequenced with Illumina machine. Mean-average plots were plotted to show the upregulated and downregulated repeats in ATRT samples vs SMARCB1-wildtype tumor samples. CPM and TPM normalizations were calculated for all repeat elements. Genomic annotation for upregulated repeat elements was performed to evaluate the genomic distribution of the upregulated repeats elements. The counts of the upregulated repeat elements that were annotated to each genomic feature were calculated. These counts were compared with the counts of genomic annotation of all repeat elements in the whole genome using the Fisher exact test to calculate the odds ratio and p-value. The reads for the genes were counted and the TPM expression values of the genes were calculated in each RNA-seq patient sample.

Differential analysis for SMARCB1-negative ($n = 3$) versus SMARCB1-positive ($n = 3$) I2A cell line samples (2 days post the re-expression of SMARCB1) was performed to evaluate the upregulated and downregulated repeats. Upregulated repeats were determined using the $\log_2$(FC) > 1 and FDR < 0.05 cut-offs. Downregulated repeats were determined using the $\log_2$(FC) < 1 and FDR < 0.05 cut-offs.

### Expressed IR-Alus analysis
Alu elements in each sample that have expression of CPM value >5 have been determined to be expressed elements. The expressed Alus in each sample have been intersected with the list of annotated IR Alus[36] to get the list of IR Alus that are expressed in each sample. We made a list of the union of these expressed IR Alu elements in all the samples. We calculated the TPM scores for the union list in each sample as the sum of the TPM values of that list of expressed IR Alu in each sample.

### ISG score
ISG score was calculated for 38 ISG genes[39] by normalizing the expression of each gene in all samples by the mean of expression of the gene in all samples to remove the bias in the baseline expression of each of the 38 genes. The ISG score in each sample is calculated as the mean of the normalized TPM values of the 38 genes in each sample.

### CTY score
We performed the quantile normalization of the TPM values after performing the $\log_2$(TPM + 1) normalization for these TPM values. The geometric mean was calculated for the PRF1 and GZMA genes to calculate the CTY score in each sample as previously described[38].

### CUT&RUN and data analysis
BT16 cells were treated with UNC1999 or UNC2400 at 5 µM and harvested at day 4 and day 6. CUT&RUN was performed as previously described[37]. Briefly, 250,000 cells from each condition were immobilized on magnetic beads, and were then incubated with H3K27me3

antibody (anti-histone H3 (tri methyl K27) antibody-ChIP grade (ab6002)) or H3K4me3 antibody (anti-histone H3 (tri methyl K4) antibody-ChIP grade (ab8580)) at 4 °C at 1:100 dilution overnight. Digitonin wash buffer (0.05%) was applied to cells the next way, followed by addition of approximately 700 ng µl − 1 protein A-MNase into the mixture. After the 1 h of rotation at 4 °C, the mixture was washed with digitonin wash buffer (0.05%). To initiate the cleavage step, 3 µl of 100 mM Ca2+ was added to the mixture on ice for 30 min. Then, chelation was used to stop the cleavage step. Following centrifugation, the histone-DNA complexes in the supernatant were then extracted using QIAGEN MinElute Kit. The extracted DNA was subjected to library preparation. NEBNext Ultra II DNA Library Prep Kit was used to prepare paired-end sequencing libraries by following the instructions of the manufacturer. Approximately 40 million reads per sample were achieved via Illumina Nextseq500 instrument sequencing.

Fastq reads were trimmed using fastp program v.0.19.5[68] to remove adapters and remove bases with low quality score. Bases with quality score less than 30 were removed, and trimmed reads that are less than 35 bp were removed. Trimmed reads were aligned to the human genome hg38 using bowtie2 tool v.2.3.5[69] using the same alignment setting described previously[70]. Duplicated reads were kept in the downstream analysis, while the unaligned reads and discordantly aligned reads were discarded. We further filtered reads that were aligned to the hg38 genome by removing reads that overlap with the hg38 ENOCDE blacklist regions (https://www.encodeproject.org/files/ENCFF356LFX/) to remove potential blacklist regions. These reads that overlap with blacklist regions were filtered using pairtobed command from the bedtools suite v.2.27.1[71].

The trimmed reads were also aligned to the S. cerevisiae yeast genome (sacCer3) to quantify the yeast spike-in reads for the spike-in normalization purpose using the bowtie2 parameters setting of alignment to the spike-in genome described previously[70]. Alignment files were converted from sam to bam format and sorted by genomic coordinates using samtools v.1.9[62].

Alignment files of biological replicates were pooled using samtools. Whole genome coverage tracks in bigwig file format were created from the pooled bam files using the bamCoverage command from the deeptools package v.3.5.0[63]. In calculating the genome tracks, we counted the spike-in reads that were aligned to the yeast genome to calculate the spike-in normalization factor that was used to normalize read counts with the bamCoverage command. Normalization factor was calculated as 10000 divided by the count of spike-in reads. Heatmaps of the CUT&RUN signal were extracted from the bigwig files using computeMatrix command and plotted using plotHeatmap command in the deepTools package.

### IR-Alu analysis
To analyze the IR-Alu elements using the hg38 genome reference and based on the experimentally-validated (EV) IR-Alu elements that are MDA5 dsRNA agonists identified previously[36], we re-analyzed the MDA5 protection assay based on the hg38 genome reference using the method described previously[36]. We detected about 1038 IR pairs which include 1587 Alu elements as baseline IR-Alu elements, and 3719 IR pairs which include about 5261 Alu elements as treatment-induced IR-Alu elements. These numbers are very close to the numbers previously detected based on hg19 genome[36]. We also performed whole-genome IR-Alu analysis to annotate all possible IR-Alu pairs in the hg38 genome. We detected about 750,005 annotated IR pairs, which is very close to the number of pairs detected previously based on the hg19 reference genome. The genomic regions of the IR-Alu elements were annotated using GENCODE v.28 assembly. Alu elements that are not overlapping with introns, exons, 3′UTR and 5′UTR regions were considered as Intergenic Alu elements.

We determined the upregulated EV and Annotated IR pairs in ATRT cells in UNC1999 versus UNC2400 samples as the pairs where

both Alu elements have $\log_2FC > 1$ and FDR < 0.05 after preforming differential analysis for the whole genome repeat elements using edgeR package. We detected the set of genes that overlap with intronic and 3′UTR IR pairs, which are the IR pairs that annotated to intronic or 3′UTR regions. We called these IRs intronic IRs or 3′UTR IRs and combined them as intragenic IRs. Pathway analysis was performed on the upregulated genes that overlap with intragenic IRs using the R package gprofiler2 to identify the enriched pathways and the transcription factor enrichment at these genes.

Heatmaps for the RNA-seq signal of genes with upregulated intronic annotated IRs were plotted from the transcription start site (TSS) to the transcription end site (TES) of these genes with expanding the plot by 1 kb upstream of the TSS and downstream of the TES. RNA-seq signal for each gene was plotted from the stand where the gene is transcribed. Heatmap data matrix was computed using the compute-Matrix command from deepTools package and plotted using plotHeatmap from the same package. Another heatmap showing the CUT&RUN H3K27me3 signal at these genes was also plotted from the TSS to the TES with expanding the plot by 2 kb upstream of the TSS and downstream of the TES.

Another heatmap of the RNA-seq signal for these overlapping genes was plotted only at the intronic regions of the genes. The tracks of this heatmap were plotted from the 5′end of the first intron to the 3′ end of the last intron in the gene. To plot this heatmap, RNA-seq signal for each intron of a gene was extracted from the bigwig file of the strand where the gene is transcribed using the pyBigWig v.0.2.7 python package. RNA-seq signals of the introns of the same genes were concatenated with keeping the order of the introns from the 5′end to the 3′ end. If the gene is transcribed in the reverse strand, the concatenated signal of the gene is reversed such that the 5′end of the signal will always be at the left side. The concatenated signal of each gene was normalized to be the same length for all genes. Signals for the genes from all plotted samples were stored in a data matrix such that each row represents a gene signal with the first group of columns representing signal from the first sample and the second group of columns representing signal from the second sample. Signal matrix was converted to a format compatible with plotHeatmap in deeptools. This matrix was used by the plotHeatmap command from the deepTools package to plot the heatmap.

Correlation scatter plots were plotted for $\log_2FC$ of the genes that overlap with upregulated intragenic annotated IRs and the $\log_2FC$ of these IRs. $\log_2FC$ of the upregulated IR was calculated as the average of the $\log_2FC$ of the two Alu elements. For genes overlapping with multiple IR pairs, we selected the IR pair with the highest $\log_2FC$ for calculating the correlation. The scatter plot was plotted using the R package ggplot2. Pearson correlation coefficient was calculated with a p-value using two-sided t test.

Correlation scatter plots were plotted for genes with upregulated IR-Alu elements in their 3′UTR and intronic regions. These scatter plots show the correlation between the $\log_2FC$ of RNA-seq and the $\log_2FC$ of each of the H3K4me3 and the H3K27me3 marks, and between the two histone marks. The FC of the RNA-seq of the genes was calculated as the ratio between the Count per Million (CPM) of the gene at the UNC1999 condition to the CPM of the gene at the UNC2400 condition. The FC of each of the two histone marks was calculated as the ratio of the average of the genomic signal of the histone mark from 5 kb upstream the TSS to 5KB downstream the TSS of the gene in the UNC1999 condition to the average of the same region in the UNC2400 condition. Histone mark signal was extracted from the bigwig file using the pyBigWig v.0.2.7 python package. Pearson correlation coefficient was calculated, and the p-value was calculated using the t test. Gene set enrichment analysis (GSEA) was performed on upregulated genes overlapping with upregulated intragenic annotated IRs. GSEA tool v.4.2.3[72,73] was used in this analysis.

## Full-length LINE-1 analysis

Upregulated LINE-1 elements were classified as full-length LINE-1 elements that have a length >= 6 kb and truncated LINE-1 elements that have a length less than 6 Kb. Donut plots for the proportions of upregulated full-length and truncated LINE-1 elements were generated. Counts of these two groups of upregulated LINE-1 elements were compared with the counts of full-length and truncated LINE-1 elements in the whole genome using the two-sided Fisher exact test to calculate the odds ratio and p-value.

Correlation scatter plots were plotted for the RNA-seq FC of the upregulated full-length LINE-1 elements and the FC of CUT&RUN histone marks. These scatter plots included RNA-seq $\log_2FC$ versus H3K4me3 $\log_2FC$, RNA-seq $\log_2FC$ versus H3K27me3 $\log_2FC$, and H3K4me3 $\log_2FC$ versus H3K27me3 FC. The FC of the H3K4me3 and H3K27me3 marks was calculated as the ratio of the CUT&RUN signal in the UNC1999 sample to the UN2400 sample at 5 kb upstream of the 5′UTR and 5 kb downstream of the 5′UTR of the elements.

## L1H and L1P sequence analysis

Full length DNA sequences of L1PA2, L1PA3, L1PA4, L1PA5, L1PA6 and L1PA7 elements have been aligned individually to a sequence of L1H. The pairs of sequences were aligned in using the Needleman-Wunsch global alignment algorithm. The following score settings were used in alignment: 1 penalty score for gap extension, 10 penalty score for gap opening, 1 score for match, and 0 score for mismatch. The alignment data has been plotted as genome tracks with color-coding the gaps in L1PA and L1H, mismatches, matches, location of 5′ UTR guides, and location of ORF1 guides.

## 5′ UTR and ORF1 guide search

We searched for the presence sequences of 5′ UTR and ORF1 guides in all LINE-1 elements in the whole genome. The counts of each LINE-1 subfamily that included the guide were compared with the total count of the corresponding subfamily in the whole genome using the two-sided Fisher exact test to calculate the odds ration and p-value. Donut plots were plotted to visualize the proportions of the LINE-1 subfamilies that included the sequences of the guides.

## Plasmids and lentiviral transduction

sgRNA sequences listed in Supplementary Table 1 targeting MAVS, cGAS, TP53, Luciferase and a control region (AAVS locus) were cloned into the lentiCRISPRv2 plasmid (gift from Feng Zhang - Addgene 52961 and 83480) as previously described[74]. Lentiviral particles were generated using HEK293FT cells as previously described[75], and the packaging plasmids pMDG.2 and psPAX2 (gifts from Didier Trono - Addgene 12259 and 12260). Cells were transduced for 24hrs followed by selection with media containing puromycin (2 µg/mL) or blasticidin (5 µg /mL) for 72hrs. Cells were washed with PBS and seeded for experimental procedures.

## RT−qPCR

RNA was isolated from cells using RNeasy Mini Kit (QIAGEN) with DNase I treatment, and quantified by a NanoDrop 2000 spectrophotometer (Thermo Scientific). SuperScript VILO cDNA Synthesis Kit (Thermo Fisher) was used to synthesize cDNA from 500 ng to 1 µg of RNA. Then, qPCR was performed in duplicates using 10 to 20 ng of cDNA. Each qPCR reaction used 5 µl of cDNA, 10 µl of SYBR Select Master Mix (Thermo Fisher) and 0.2 µM of forward and reverse primers in 20 µl total volume, which was performed by StepOnePlus Real-Time PCR system (Applied Biosystems). The condition of the qPCR reaction was: 95 °C for 10 min, 40 cycles of 95 °C for 15 s, and 60 °C for 1 min. The expression levels of the target gene were normalized against the housekeeping gene RPLP0 and referred to control using the formula ($2^{-\Delta\Delta CT}$). The Supplementary Table 2 shows the sequences of the primers used.

## Immunocytochemistry

Cells were grown on glass coverslips and fixed for 20 min in 4% paraformaldehyde (PFA), permeabilized for 5 min with 0.5% Triton X-100, blocked with blocking buffer (BSA 3%) for 1 h at 37 °C and incubated with primary antibodies diluted in 3% BSA at 4 °C overnight. The primary antibodies for immunocytochemistry were shown in Supplementary Table 3. On the next day, secondary antibodies including anti-mouse IgG (H + L) (Cell Signaling Technology, 4410) and Anti-rabbit IgG (H + L) (Cell Signaling Technology, 4412) were used at a dilution of 1:1000 to incubate the cells for 1 h at room temperature. Cells were then incubated with DAPI before mounting with ProLong Gold Antifade Mountant (Thermo Fisher Scientific, P36930). Confocal microscopy was performed using Leica SP8 Confocal Microscope with consistent acquisition parameters. captured images were analyzed with FIJI (ImageJ). Corrected total cell fluorescence (CTCF) was calculated as: integrated density − (area of selected cell × mean fluorescence of background readings). The number of γH2A.X foci per cell was quantified by manually counting γH2A.X foci under blinded conditions. Micronuclei were characterized as discrete DNA structures detached from the primary nucleus, while apoptotic appearance cells were excluded from the analysis. The fraction of cells containing micronuclei was determined via microscopy in blinded conditions with DAPI staining.

## ssDNA staining

ssDNA immunocytochemistry was conducted as previously described[56]. Briefly, cells were fixed with 4% PFA on ice for 20 min and then were incubated with methanol overnight at −20 °C. The following day, 200 µg/mL RNase (Sigma R4642) was applied on the cells for 4 hours at 37 °C. 3% BSA was then used to block the cells for 1 h. Cells were incubated with the anti-ssDNA primary antibody (Millipore, MAB3299, 5 µg/mL) at 4 °C overnight. On the next day, the secondary antibodies and then DAPI were used to incubate the cells prior to mounting. Leica SP8 Confocal Microscope was used to blindly obtain images of ssDNA. The number of ssDNA puncta per cell was determined by manually counting ssDNA puncta under blinded conditions.

## MAVS aggregation assay

MAVS aggregation was examined using semi-denaturing detergent agarose gel electrophoresis (SDD-AGE) gel as previously described[76].In summary, mitochondria were isolated (Qproteome Mitochondria Isolation Kit, Qiagen) from BT16 and CHLA02 cells after casting a 1.5% agarose vertical gel. The isolated mitochondria were resuspended in mitochondria buffer, and diluted before loading on the gel. Electrophoresis was done for 1 h at 4 °C at a constant voltage of 100 V in running buffer (1× TBE and 0.1% SDS). Then, proteins were transferred into a Trans-Blot Turbo Midi PVDF Transfer membrane (BioRad), and MAVS protein was detected using anti-MAVS antibody in 1:1,000 dilution (ab89825; Abcam).

## Western blot

Cells were lysed in lysis buffer (0.1% SDS, 400 mM NaCl, 1 mM EDTA, 50 mM of Tris-HCl, 1% Triton) and applied haltprotease and phosphatase inhibitor cocktail (Thermo Fisher Scientific) as the final dilution of 1:100 to obtain whole-cell extracts. A BCA assay (Promega) was then used to quantify protein. After boiling at 95 °C for 5 min, 20–30 µg of protein per lane was loaded and separated on 4–20% Mini-PROTEAN TGX Gel (BioRad). Then, protein was transferred onto a Trans-Blot Turbo Nitrocellulose Transfer membrane (BioRad). The membrane was blocked in 5% BSA in TBS + 0.1% Tween 20 (TBST) and incubated in primary antibody diluted in blocking buffer overnight at 4 °C. The primary antibodies for western blot were listed in Supplementary Table 3. The following secondary antibodies were used: Anti-Rabbit IgG, HRP-linked Antibody (Cell Signaling, 7074, 1:5000) and anti-Mouse IgG, HRP-linked Antibody (Cell Signaling, 7076, 1:5,000). ClarityWestern ECL Substrate (BioRad) was used to develop the blots.

## Isolation of ssDNA species

Extrachromosomal DNA extraction and ssDNA preparation was conducted as previously described[56]. In brief, BT16 cells were detached using trypsin, pelleted, and suspended in Buffer 1 (50 mM Tris-HCl, 10 mM EDTA, supplemented with 100 µg/mL RNase A). Subsequently, cells were lysed with Buffer 2 (1.2% sodium-dodecyl sulfate), ensuring complete lysis through a 5-minute incubation at room temperature. Debris and high-molecular-weight chromosomal DNA were precipitated by adding Buffer 3 (3 M CsCl, 1 M potassium acetate, and 0.67 M acetic acid). After chilling on ice for 15 minutes, the supernatant containing extrachromosomal DNA was collected, column purified, and treated with dsDNase (ThermoScientific) before a second column purification to obtain ssDNA. L1 enrichment in ssDNA were quantified by qPCR using primers targeting 5′UTR, ORF1, and ORF2 regions, with normalization to the corresponding gDNA. The primers used for qPCR were included in Supplementary Table 2.

## dsRNA immunoprecipitation

The dsRNA immunoprecipitation was adapted from previously established method[77]. One day before harvesting the cells, Protein G Dynabeads (Thermo 10004D) were washed three times with antibody conjugation buffer before being resuspended beads in antibody conjugation buffer (1x PBS, 2 mM EDTA,0.1% BSA). In all, 5 µg J2 anti-dsRNA antibody (Scicons) per reaction was added to the washed beads and incubated overnight at 4 °C with rotation. BT16 cells were detached by trypsin and resuspended in lysis buffer (20 mM Tris pH 7.5, 150 mM NaCl, 10 mM EDTA, 10% Glycerol, 0.1% NP-040, 0.5% Triton X-100, protease inhibitors (add fresh)) at day 4 post UNC1999 or UNC2400 treatment. Cell lysates were incubated for 15 minutes on ice. Cell debris was cleared via 15 min max speed centrifugation. 10% of the cleared cell lysate was retained as input control. Antibody-bound beads were then washed three times with ice-cold lysis buffer before being added to the IP fraction. The mixture was incubated overnight at 4 °C with rotation. RNA-bound beads were then washed three times with ice-cold high salt wash buffer before RNA extraction using TRIzol (along with input). Input and dsRNA-enriched fractions were then converted into cDNA and analyzed by qPCR. The primers used for qPCR were included in Supplementary Table 2.

## Statistical analysis and illustrations

The sample size in each experiment is indicated in the figure legends. Statistical analyses and plotting for in vitro assays were performed using GraphPad prism (GraphPad Software, USA). Schematic illustrations were created using BioRender.

## Reporting summary

Further information on research design is available in the Nature Portfolio Reporting Summary linked to this article.

# Data availability

Raw sequencing data of RNA-seq and CUT&RUN have been deposited at the Gene Expression Omnibus (GEO) (https://www.ncbi.nlm.nih.gov/geo) under the accession number GSE213250. Source data are provided with this paper.

# Code availability

Custom codes used for data analysis are available at the project Zenodo repository (https://zenodo.org/records/10534910).

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

## Acknowledgements

We thank members of the De Carvalho laboratory for their discussions and input. We would like to thank Princess Margaret Cancer Centre Genomics core (PM Genomics) for library preparation and sequencing on RNA-seq and CUT&RUN samples. This work was supported by CIHR Foundation Grants (FDN 148430), CIHR Project Grant (PJT 165986), NSERC (489073), Canada Research Chair, New Investigator salary award (201512MSH360794-228629), Helen M Cooke professorship from Princess Margaret Cancer Foundation, and Ontario Institute for Cancer Research (OICR) with funds from the province of Ontario to D.D.D.C., CIHR (FDN154328) to C.H.A., NSERC (RGPIN-2019-07041) to M.D.W. and D.J.S., NSERC PGS D to D.J.S., and National Natural Science Foundation of China (No. 32400487) to S.F. The Structural Genomics Consortium (SGC) is a registered charity (no: 1097737) that receives funds from Bayer AG, Boehringer Ingelheim, Bristol Myers Squibb, Genentech, Genome Canada through Ontario Genomics Institute [OGI-196], EU/EFPIA/OICR/McGill/KTH/Diamond Innovative Medicines Initiative 2 Joint Undertaking [EUbOPEN grant 875510], Janssen, Merck KGaA (aka EMD in Canada and US), Pfizer and Takeda.

## Author contributions

Conceptualization: S.F. and D.D.D.C. Methodology and experiments: S.F., F.S., P.M., C.I., H.L.Y., I.E., P.S.P., R.C., J.L., P.C.Z., K.C.H., B.H., and S.N. Data analysis: S.A.M., D.J.S., A.D.C., and S.F. Original draft: S.F., S.A.M., D.J.S., C.H.A., and D.D.D.C. Supervision: A.H., M.D.W., J.T.S., C.H.A., and D.D.D.C.

## Competing interests

D.D.C. reports grants from Princess Margaret Cancer Foundation, the Canadian Institutes of Health Research, and Canada Research Chair during the conduct of the study; grants from Pfizer, and other support from Adela, Inc outside the submitted work. All the other authors declare no conflict of interest.
