## [Peer Review File · Nature Communications]

Inhibiting EZH2 targets atypical teratoid rhabdoid tumor by triggering viral mimicry via both RNA and DNA sensing pathwaysREVIEWER COMMENTS

Reviewer #1 (Remarks to the Author):

The manuscript submitted from Feng et al. investigates down stream mechanisms of action following EZH2 inhibition in Atypical Teratoid Rhabdoid Tumors (ATRT). ATRT are tumors with inactivating mutations in SMARCB1 which denotes tumors dependent on EZH2 oncogenic function. However, little is known regarding downstream mechanisms following EZH2 inhibition.

Within, the authors nicely demonstrate that the EZH1/2 inhibitor - UNC1999 significantly remodels chromatin and alters gene expression. The team describes that EZH1/2 inhibition induces viral mimicry by both increased expression of dsRNA and dsDNA due to intronic inverted repeat Alu (IR-Alu) and LINE-1 elements respectively. This data is extremely timely and an important to the overall field of cancers with SWI/SNF LOF mutations where EZH2 dependency seems universal. As presented, the current manuscript does need experimental additions to elevate its translation and overall impact for publication in Nature Communications.

1) It is felt from this reviewer that it would be first beneficial to determine if increased DNA damage and expression of ISGs (viral mimicry) results in cell death or not, and if this response is related to overall immunogenicity of the tumor cell. Translating this knowledge to therapy response to checkpoint inhibition should also be demonstrated (in vitro or in vivo) if increased tumor cell immunogenicity is found.

2) Also, it would be important to extend these cell line results into patient datasets.

3) Minor - UNC1999 is a EZH1/2 inhibitor - does a more clinically relevant EZH1/2 inhibitor (Valemetostat) produce similar results?

Reviewer #2 (Remarks to the Author):

The article by Feng et al describes the effect of EZH2 inhibition by pharmacological inhibitors in SMARCB1-deficient cell lines. They showed some growth impairment on the four cell lines and that EZH2 inhibition resulted in the induction of interferon-signaling and cytokine, in a JAK pathway dependent manner.

Then, they hypothesized that ISG were induced by dsRNA sensing pathways. They find that dsRNA was indeed increased in ATRT cell lines at day after UNC1999 treatment.

They inactivated MAVS in one cell line BT16 by CRISP-Cas9 and found that ISG expression was partially reduced. They analyzed the level of expression of all repeated elements, and indeed found over-expression of all classes, among which IR-Alu were the most significant. They could relate this induced expression to H3K27me3 depletion and H3K4me3 increase mostly inducing intronic IR-Alu but not intergenic IR-ALu.

The authors then observe that induced expression of intronic IR-Alu correlated with an induced expression of their associated gene; ISG were found among such genes.

The authors then show that the induction of interferon signaling pathway is not fully inactivated by MAVS KO, and that cCAS-STING is also playing a significant, if not more important, role. They observe that gH2AX foci are induced upon EZH2 treatment, and that micronuclei are generated. An induction of the LINE-dependent retro-transposition machinery is at the origin of an important genome instability that senses the dsDNA mediated cGAS-STING pathway.

The article is adding new mechanisms in the field and is definitely of broad interest for many cancer types.

A few questions nevertheless remain unanswered

-a pharmacological inhibition may have some off-target effects; the use of UNC1999/UNC2400

matched compound limits the risk of over-interpretation of off-target effects, and the authors have verified that another pharmacological inhibitor does similar effects; however, it would be even nicer to check whether a genetic depletion of EZH2 inhibition provides similar effects and at similar levels.

-the authors show a rather toxic effect of UNC1999 on all cell lines at D14; all the subsequent analyses are performed at earlier time points (D3/7 or 4/6), presumably to avoid the background noise of apoptosis at later time points; however, EZH2 inhibition is meant to be given during longer time in patients, and whether viral mimicry is actually relevant in a long-time exposure is critical for clinical relevance; it would be interesting to analyse whether the viral mimicry is an actual important mechanisms after longer exposure, this may be critical to predict what may be going on in the clinics; in order to avoid the possible biases at D14 due to other toxic effects, the authors could possibly expose the cells to lower doses for a longer time, in conditions still allowing cells to survive but with actual biological effect, and again check what is going on at D14 or even later.

-the relevance of using ATRT cell lines is clear from a translational perspective given the current development of EZH2 inhibitors in such diseases; however, the authors don't really develop in details to which extent the viral mimicry they depict is or not already active before treatment; could they author give an order of magnitude of how dsRNA and dsDNA sensing are active in SMARCB1-deficient cells at the basal level as compared to any relevant SMARCB1-proficient counterparts, and how this particular epigenetic background is indeed influencing the viral mimicry induction by EZH2 inh ? It could be nice to see how SMARCB1 loss is actually playing a role per se in the overall mechanisms induced by EZH inh.

-according to a Biorxiv released work by Kazansky et al, tumor genomes don't seem to be massively changed after treatment as compared to pre-treated profiles. Nevertheless, we may assume that the genomic instability is not looked at the relevant resolution; nevertheless, it would be useful to further demonstrate the extent of this genome instability by whole genome sequencing before and after treatment, at some selected time points.

-it could be interesting to have a broader overview on how H3K27me3 is actually depleted in treated cells, not only at IR-Alu or selected derepressed gene regions, but at the whole genome level;

-How to explain then the reduced expression of intergenic IR-Alu ? Is there any correlation with the depletion of H3K27me3 in intergenic regions versus intronic regions ?

-The author insist on ISG among genes with co-induced expression of IR-Alu; is there any particular biological functions for all the other genes ? What is the proportion of « non ISG » that show similar co-expression with IR-Alu and is that of particular interest to also look at those ones ?

Reviewer #3 (Remarks to the Author):

Inhibiting epigenetic regulators in tumors has the potential to activate a viral mimicry state, which stimulates antitumoral innate immunity. However, each tumor type can depend on different pathways and be vulnerable to a distinct set of epigenetic therapies. Here, the authors explore the mechanisms underlying the vulnerability of tumors deficient for the SWI/SNF complex (SMARCB1 loss-of-function) upon EZH2 inhibition, using atypical teratoid rhabdoid tumors (ATRTs) as a model. Their data confirm that EZH2i leads to a viral mimicry state as shown in other tumor types and demonstrate that this process results from the dual activation of cytosolic RNA and DNA sensing pathways. Finally, some experiments suggest that retrotransposons (Alu/L1) may trigger the activation of these pathways through different mechanisms. The study tackles fundamental questions to fully understand the mechanism of action of epigenetic drugs and will help understanding the role of endogenous retroelements in immunity. However, the link between Alu repeats and RNA sensing, as well as between L1 and DNA sensing, is preliminary and needs to be reinforced before publication.

Pros:

- new findings revealing the pathways underlying the activation of viral mimicry in tumor cells
- strong demonstration that both dsRNA and cytoplasmic DNA sensing cooperate to set up viral

mimicry upon dual inactivation of EZH2 and SMARCB1.

Cons:

- the causal relationship between Alu/L1 and viral mimicry is not proven in this context

Major points:

1. The proposed model is that Alu inverted repeats embedded in genes derepressed by EZH2i may generate structured RNA bound by MDA5, which in turn will activate MAVS and the dsRNA sensing pathway. However, the authors do not directly show here that Alu-IR are actually bound by MDA5 in the context of ATRTs. They could perform an MDA5 protection sequencing assay as achieved by the same group for demethylating agents (Mehdipour et al. Nature 2020). Also, do dsRNA foci detected by IF contain Alu sequences by RNA-FISH?

2. The link between the so-called L1 retrotransposition and cytoplasmic DNA sensing is weak in many respects.

2a. First, it appears that the expressed L1s belong to families which are unable to retrotranspose (not L1Hs) and it is unclear whether they are transcribed autonomously (from the L1 promoter) or "passively" (cotranscribed with genes in which they are embedded). A careful presentation and analysis of the full length elements assumed to be expressed is needed here. ORF1p western-blots may also reinforce the hypothesis that some intact and young L1s are expressed. Similarly, what is the extent of overlap between H3K27me3 reduction, STING activation and ORF1p expression at the single-cell level (IF)?

2b. Second, the detection of ssDNA in the cytoplasm is not evidence of retrotransposition. The authors detect cytoplasmic ssDNA upon EZH2i, and these DNA species disappear after treatment with drugs known to inhibit ORF2p reverse transcriptase activity. Nevertheless, these drugs may affect other polymerases and it still is unknown whether the ssDNA species contain L1 sequences or depend on L1 activity. FISH using L1 probes and L1 knock-down could strengthen this conclusion (see De Cecco et al. Nature 2019, Thomas et al. Cell Stem Cell 2017 for related experiments).

2c. Finally, the term "retrotransposition" is misemployed. "L1 retrotransposition" refers to the entire cycle of L1 replication, which includes expression, assembly of L1 RNPs, and target-primed reverse transcription, ultimately leading to new insertions. I would suggest the authors substitute this term by "L1 reverse transcription" or "L1 enzymatic activities" in the context of their experiments.

Minor points:

3. What is the % of H3K27me3-bound Alu-IR that are upregulated, and how does it compare to unbound Alu-IR ?

4. Refs 4 and 18 appear to be duplicates.

REVIEWER COMMENTS

Reviewer #1 (Remarks to the Author):

The manuscript submitted from Feng et al. investigates down stream mechanisms of action following EZH2 inhibition in Atypical Teratoid Rhabdoid Tumors (ATRT). ATRT are tumors with inactivating mutations in SMARCB1 which denotes tumors dependent on EZH2 oncogenic function. However, little is known regarding downstream mechanisms following EZH2 inhibition.

Within, the authors nicely demonstrate that the EZH1/2 inhibitor - UNC1999 significantly remodels chromatin and alters gene expression. The team describes that EZH1/2 inhibition induces viral mimicry by both increased expression of dsRNA and dsDNA due to intronic inverted repeat Alu (IR-Alu) and LINE-1 elements respectively. This data is extremely timely and an important to the overall field of cancers with SWI/SNF LOF mutations where EZH2 dependency seems universal. As presented, the current manuscript does need experimental additions to elevate its translation and overall impact for publication in Nature Communications.

We thank the reviewer for the enthusiasm and for recognizing the impact of our study. We have conducted additional experiments and analysis to strengthen the conclusions of our manuscript. Please find detailed responses below.

1) It is felt from this reviewer that it would be first beneficial to determine if increased DNA damage and expression of ISGs (viral mimicry) results in cell death or not, and if this response is related to overall immunogenicity of the tumor cell. Translating this knowledge to therapy response to checkpoint inhibition should also be demonstrated (in vitro or in vivo) if increased tumor cell immunogenicity is found.

We thank the reviewer for this suggestion and agree it is important to evaluate the impact of EZH2 inhibition on cell death. To address the reviewer's comment, we conducted Annexin V and propidium iodide (PI) labeling, followed by flow cytometry. Analysis of apoptosis in CHLA02 cells revealed a dose-dependent increase in apoptotic fractions (Annexin V-positive) following UNC1999 treatment, concomitant with a decrease in the percentage of viable cells (Annexin V and PI-negative) (New **Extended Data Fig. 1d, e, f**). Moreover, this effect on cell death induction by UNC1999 was consistently observed across all tested ATRT cell lines (New **Extended Data Fig. 1e, f**). In summary, our analyses of cell proliferation and apoptosis collectively suggest that impaired cell proliferation and increased apoptosis contribute to the growth inhibitory effects of EZH2 inhibition in ATRT cells.

Extended Data Fig. 1 d, Representative flow plots of Annexin V/PI staining in CHLA02 cells treated with DMSO, UNC1999, or UNC2400 from four independent experiments. **e,f**, Quantification of apoptotic (AV+) fractions (**e**) and viable cell fractions (AV-/PI-)(**f**) from four independent experiments. Error bars represent the SD of four independent experiments. * $p < 0.05$; ** $p < 0.01$; *** $p < 0.001$ (t test with Welch's correction).

Triggering innate immunity in cancer cells via cytoplasmic nucleic acid sensing pathways, termed "viral mimicry," has shown efficacy in transforming immunologically "cold" tumors into "hot"¹⁻⁴. This response enhances tumor immunogenicity by increasing antigen presentation, including peptides derived from retrotransposons, which may serve as tumor-associated antigens and stimulate the adaptive immune response^{5,6}. Notably, epigenetic therapies, such as EZH2 inhibition, elicit interferon signaling, leading to the transcriptional activation of antigen presentation machinery (e.g., MHC-I, $\beta 2m$) through a broad spectrum of interferon-stimulated genes⁷⁻¹¹. The activated interferon signaling and enhanced processing and presentation of the retrotransposon-derived peptides contribute to elevated tumor immunogenicity, providing a potential rationale for the positive correlations between increased retrotransposon expression, augmented T-cell infiltration, and improved response to immune checkpoint blockade (ICB) therapy^{3,9,11-15}. More recently, Morel et al. showcased that inhibiting EZH2 in prostate cancer

models triggers a dsRNA–STING–ISG stress response, leading to the upregulation of genes associated with antigen presentation, Th1 chemokine signaling, and interferon response¹¹. This includes the expression of programmed cell death protein 1 (PD-L1), contingent upon STING activation. In line with these findings, we observed a significant upregulation of CD274 (PD-L1) and genes involved in antigen presentation (e.g., HLA-A, HLA-B, HLA-C, B2M) upon UNC1999 treatment (**Reviewer Figure 1**), suggesting that EZH2 inhibition may enhance tumor immunogenicity and potentially promote sensitivity to ICB therapy in ATRTs.

Reviewer Figure 1. Error bar plots showing the Count Per Million (CPM) expression scores of the HLA-A, HLA-B, HLA-C, B2M and CD274 genes from left to right in samples treated with UNC2400 or UNC1999 at day6. P-value was calculated using one-tailed paired t test. * $p < 0.05$, ** $p < 0.01$ and *** $p < 0.001$.

2) Also, it would be important to extend these cell line results into patient datasets.

We appreciate the reviewer's insightful comment and constructive suggestion to extend our analysis from cell line results to patient datasets. In response, we broadened our investigation to include an RNA-seq dataset of primary brain tumors, encompassing both SMARCB1-wildtype brain tumors and SMARCB1-deficient ATRTs. Despite a low tumor mutational burden, rhabdoid tumors including ATRTs demonstrate significant immune infiltration¹⁵. The elevated immunogenicity observed in rhabdoid tumors implies a potential elevation in the baseline expression of retroelements, which may prime these tumors for viral mimicry induction. Indeed, differential analysis of repeat expression revealed that more retroelements exhibited higher expression in ATRTs compared to SMARCB1-wildtype brain tumors, with SINE elements being particularly overrepresented among the upregulated repeats (New **Extended Data Fig. 5a, b**). Notably, Alu elements showed significant enrichment among the upregulated repeat families in ATRTs (New **Extended Data Fig. 5a, b**). Further scrutiny of the expression of annotated IR-Alu elements revealed a substantially greater number of IR-Alu repeats with increased expression levels in ATRTs when compared to SMARCB1-wildtype intracranial tumors (New **Extended Data Fig. 5c, d**). A significant proportion of these repeats exhibited upregulation on both strands (New **Extended Data Fig. 5e**), suggesting that the elevated expression of IR-Alu pairs in ATRTs may lead to the formation of dsRNA. Additionally, the total expression of IR-Alu repeats was computed, demonstrating an overall increased expression in ATRTs compared to SMARCB1-wildtype brain tumors (New **Extended Data Fig. 5f**). Collectively, these findings suggest that the loss of SMARCB1

may prime ATRTs for the induction of viral mimicry, a response that can be further enhanced with EZH2 inhibition.

Extended Data Fig. 5. a, Mean-average (MA) plots showing the upregulated and downregulated repeats in ATRT samples (n=11) versus SMARCB1-wildtype tumors (n=14) for all repeat, SINE and Alu elements. The x axis depicts the average of log(CPM) in all samples and the y axis depicts log(FC). The red dots represents the upregulated repeat element which have log(FC)>1 and FDR<0.05, while the blue dots represent the downregulated repeat elements which have log(FC)<1 and FDR<0.05. Gray dots represent the repeat elements that are not significantly regulated. **b**, Donut plots showing the percentages of repeat classes (left) and major families (right) that are upregulated in ATRTs when compared with SMARCB1-wildtype tumors. Odds ratio and p-value are calculated using Fisher exact test by comparing the counts of the elements of these upregulated classes/families compared to the expected genomic distribution. ****p<0.0001.

Extended Data Fig. 5. c, Volcano plot showing log₁₀(FDR) versus log(FC) of upregulated (red dots) and downregulated (blue dots) IR-Alus in ATRTs (n=11) compared with SMARCB1-wildtype tumors (n=14). **d**, Scatter plot showing the average of log(CPM) values of the upregulated (red dots) and

downregulated (blue dots) IR-Alus in ATRT samples versus SMARCB1-wildtype tumors. **e**, Scatter plot showing the $\log(\text{FC})$ for both Alus in upregulated/downregulated IR-Alu pairs. The x axis depicts the $\log(\text{FC})$ of one Alu, while the y axis depicts the $\log(\text{FC})$ of the other Alu. An IR-Alu pair is considered differentially regulated when both Alus exhibit $|\log(\text{FC})| > 1$ and $\text{FDR} < 0.05$.

f

Extended Data Fig. 5. f, Violin plot showing the sum of the TPM values of the union list of IR-Alus with $\text{CPM} \geq 5$ in each sample of the ATRTs versus SMARCB1-wildtype tumors. The p -value was calculated using the Wilcoxon ranked sum test.

3) Minor - UNC1999 is a EZH1/2 inhibitor - does a more clinically relevant EZH1/2 inhibitor (Valemetostat) produce similar results?

We appreciate the reviewer's feedback. In response, we extended our investigation to include two additional clinically relevant inhibitors: Valemetostat (DS-3201, an EZH1/2 inhibitor) and Tazemetostat (EPZ6438, an EZH2 inhibitor). Notably, both inhibitors demonstrated a significant induction of interferon-stimulated genes at both day 6 and day 12 following treatment (New **Extended Data Fig. 2e, f**). These findings provide additional robust evidence supporting the role of EZH2 inhibition in promoting viral mimicry in ATRT cancer cells within a more clinically relevant context.

Extended Data Fig. 2. e,f, The expression of indicated ISGs in BT16 cells treated with UNC2400, UNC1999, Valemetostat (DS-3201), or Tazemetostat (EPZ6438) was measured by quantitative real-time PCR at day 6 (**e**) and day 12 (**f**). Error bars represent the SD of three independent experiments. $*p < 0.05$; $**p < 0.01$; $***p < 0.001$ (two-tailed t test).

Reviewer #2 (Remarks to the Author):

The article by Feng et al describes the effect of EZH2 inhibition by pharmacological inhibitors in SMARCB1-deficient cell lines. They showed some growth impairment on the four cell lines and that EZH2 inhibition resulted in the induction of interferon-signalling and cytokine, in a JAK pathway dependent manner.

Then, they hypothesized that ISG were induced by dsRNA sensing pathways. They find that dsRNA was indeed increased in ATRT cell lines at day after UNC1999 treatment.

They inactivated MAVS in one cell line BT16 by CRISP-Cas9 and found that ISG expression was partially reduced. They analyzed the level of expression of all repeated elements, and indeed found over-expression of all classes, among which IR-Alu were the most significant. They could relate this induced expression to H3K27me3 depletion and H3K4me3 increase mostly inducing intronic IR-Alu but not intergenic IR-ALu.

The authors then observe that induced expression of intronic IR-Alu correlated with an induced expression of their associated gene; ISG were found among such genes.

The authors then show that the induction of interferon signaling pathway is not fully inactivated by MAVS KO, and that cCAS-STING is also playing a significant, if not more important role. They observe that gH2AX foci are induced upon EZH2 treatment, and that micronuclei are generated. An induction of the LINE-dependent retro-transposition machinery is at the origin of an important genome instability that senses the dsDNA mediated cGAS-STING pathway.

The article is adding new mechanisms in the field and is definitely of broad interest for many cancer types.

A few questions nevertheless remain unanswered.

We thank the reviewer for the positive feedback and the constructive criticism. We addressed each one individually below. These suggestions have greatly improved our revised manuscript.

1. a pharmacological inhibition may have some off-target effects; the use of UNC1999/UNC2400 matched compound limits the risk of over-interpretation of off-target effects, and the authors have verified that another pharmacological inhibitor does similar effects; however, it would be even nicer to check whether a genetic depletion of EZH2 inhibition provides similar effects and at similar levels.

We appreciate the reviewer's valuable suggestion to explore the effects of genetic inactivation of EZH2. In addition to the data on UNC1999 and GSK343 presented in our initial manuscript, we have incorporated findings from two clinically relevant EZH2 inhibitors, Valemetostat and Tazemetostat. The results demonstrate a similar induction of viral mimicry response (New **Extended Data Fig. 2e, f**) (See Reviewer #1 comment #3 above). Furthermore, to strengthen our evidence through genetic inactivation, we conducted EZH2 knockout experiments in the ATRT

cell line BT16. Our findings indicate significant growth inhibition in EZH2-deficient cells (New **Extended Data Fig. 2g, i**). Additionally, a notable increase in interferon-stimulated gene expression was observed upon EZH2 knockout in BT16 cells (New **Extended Data Fig. 2h**). Collectively, these data from both pharmacological and genetic inhibition approaches demonstrate that EZH2 inhibition triggers a viral mimicry response in ATRT cancer cells.

Extended Data Fig. 2. g, Representative immunoblots showing the expression level of EZH2 and Vinculin in BT16 cell line with sgRNA against negative control and EZH2. **h**, The expression of indicated ISGs in BT16 cell line with sgRNA against negative control and EZH2 was measured by quantitative real-time. sgEZH2 #1 and sgEZH2 #1 represent two independent knockout cell lines deficient for EZH2. Error bars represent the SD of three independent experiments. *p < 0.05; **p < 0.01; ***p < 0.001 (two-tailed t test). **i**, Number of viable cells in culture of BT16 cell line with sgRNA against negative control and EZH2 were counted at day 0, day 4, and day 6. Error bars represent the SD of three independent experiments. ***p < 0.001 (two-tailed t test).

2. the authors show a rather toxic effect of UNC1999 on all cell lines at D14; all the subsequent analyses are performed at earlier time points (D3/7 or 4/6), presumably to avoid the background noise of apoptosis at later time points; however, EZH2 inhibition is meant to be given during longer time in patients, and whether viral mimicry is actually relevant in a long-time exposure is critical for clinical relevance; it would be interesting to analyse whether the viral mimicry is an actual important mechanisms after longer exposure, this may be critical to predict what may be going on in the clinics; in order to avoid the possible biases at D14 due to other toxic effects, the authors could possibly expose the cells to lower doses for a longer time, in conditions still allowing cells to survive but with actual biological effect, and again check what is going on at D14 or even later.

We appreciate the reviewer's valuable input. In response to this suggestion, we conducted additional experiments using lower doses of UNC1999 (1 μ M or 2.5 μ M) and extended the treatment duration to 18 days in BT16 cells. We evaluated the expression of interferon-stimulated genes at day 6, day 12, and day 18. Our results indicate a significant dose-dependent increase in interferon-stimulated gene expression at day 6, with a more pronounced induction at

day 12 and persisting until day 18 (New **Extended Data Fig. 2j**). These findings suggest the clinical relevance of viral mimicry induction through prolonged exposure to EZH2 inhibition.

j

Extended Data Fig. 2. j, The expression of indicated ISGs in BT16 cells treated with UNC2400 or UNC1999 at 1 μM or 2.5 μM was measured by quantitative real-time PCR at day 6, day 12, and day 18. Error bars represent the SD of four independent experiments. * $p < 0.05$; ** $p < 0.01$; *** $p < 0.001$; **** $p < 0.0001$ (two-tailed t test).

3. the relevance of using ATRT cell lines is clear from a translational perspective given the current development of EZH2 inhibitors in such diseases; however, the authors don't really develop in details to which extent the viral mimicry they depict is or not already active before treatment; could they author give an order of magnitude of how dsRNA and dsDNA sensing are active in SMARCB1-deficient cells at the basal level as compared to any relevant SMARCB1-proficient counterparts, and how this particular epigenetic background is indeed influencing the viral mimicry induction by EZH2 inhibition? It could be nice to see how SMARCB1 loss is actually playing a role per se in the overall mechanisms induced by EZH inhibition .

We thank the reviewer for the constructive suggestion to investigate how SMARCB1 deficiency is influencing the viral mimicry per se. In response to Reviewer #1 comment #2, we have expanded our analysis to an RNA-seq dataset of primary brain tumors, covering both SMARCB1-wildtype tumors and SMARCB1-deficient ATRTs. Our results indicate that ATRTs exhibit higher expression of retroelements, particularly Alu elements, compared to SMARCB1-wildtype brain tumors (New **Extended Data Fig. 5a, b**). Analysis of annotated IR-Alu elements known to form dsRNA¹⁶, revealed a significantly greater number of these repeats with increased expression in ATRTs (New **Extended Data Fig. 5c, d**), suggesting potential dsRNA formation. The overall expression of IR-Alu repeats is higher in SMARCB1-deficient ATRTs compared to SMARCB1-wildtype brain tumors (New **Extended Data Fig. 5e**). To strengthen our findings, we re-analyzed published RNA-seq data in a rhabdoid tumors cell line (I2A) with inducible SMARCB1 re-expression¹⁵. At day 2 post SMARCB1 re-expression, we observed increased transcription of annotated IR-Alu elements in the SMARCB1-deficient condition (New **Extended Data Fig. 5g, h**). We then identified 307 IR-Alu pairs with significant overexpression at both strands in the

SMARCB1-deficient condition (New **Extended Data Fig. 5i**). In addition, we calculated the sum of the TPM of expressed IR-Alu repeats and found that re-expression of SMARCB1 led to an overall repression of IR-Alu repeats (New **Extended Data Fig. 5j**). These data suggest a baseline activation of viral mimicry by expressed IR-Alu pairs in the SMARCB1-deficient condition. Additionally, the ISG score, computed using the mean expression levels of the 38 ISGs defined by Liu et al.¹⁷, showed a significant increase in ISG expression in the SMARCB1-deficient condition (New **Extended Data Fig. 5k**). Together, our findings from both the primary brain tumor cohort and the I2A cell line suggest that the SMARCB1 deficiency may create a predisposition in ATRTs for viral mimicry induction that can be further elevated upon EZH2 inhibition.

Extended Data Fig. 5. g, Volcano plot showing $\log_{10}(\text{FDR})$ versus $\log(\text{FC})$ of upregulated and downregulated IR-Alu elements in SMARCB1 negative ($n=3$) versus SMARCB1 positive ($n=3$) I2A cell line samples at day 2. Red dots represent the upregulated IR-Alu elements which have $\log(\text{FC}) > 1$ and $\text{FDR} < 0.05$, while blue dots represent downregulated IR-Alu elements which have $\log(\text{FC}) < -1$ and $\text{FDR} < 0.05$. **h**, Scatter plot showing $\log(\text{CPM})$ of the upregulated and downregulated Alu elements in SMARCB1 negative ($n=3$) versus SMARCB1 positive samples ($n=3$). The x and y axes represent the mean of $\log(\text{CPM})$ of the elements in SMARCB1 negative and positive samples respectively. Differentially regulated IR-Alu elements are determined with $|\log(\text{FC})| > 1$ and $\text{FDR} < 0.05$. **i**, Scatter plot shows the upregulated (red dots) and downregulated (blue dots) IR-Alu pairs in SMARCB1 negative versus SMARCB1 positive samples at day 2. Gray dots represent IR-Alu pairs that are not differentially regulated. The x axis depicts the $\log(\text{FC})$ of one Alu in the pair, and the y axis depicts the $\log(\text{FC})$ of the other Alu in the pair. An IR-Alu pair is considered differentially regulated when each of the two Alus have $|\log(\text{FC})| > 1$ and $\text{FDR} < 0.05$.

*Extended Data Fig. 5. j, Violin plot showing the sum of the TPM values of the union list of the IR-Alus with CPM \geq 5 in each sample of the SMARCB1 negative versus SMARCB1 positive samples at day 2. The sum of the TPM values was calculated for the union list of IR-Alus in each sample. Lines connect the paired samples. P-value was calculated by one-tailed paired t test. * p <0.05. k, Error bar plot comparing ISG TPM scores between SMARCB1 negative and positive samples at day 2. ISG TPM scores were normalized by the mean TPM values of the gene in all samples. The score for each sample represents the mean of the normalized TPM values of 38 ISG genes. P-value calculated using one-tailed paired t-test. * p <0.05.*

4. according to a Biorxiv released work by Kazansly et al, tumor genomes don't seem to be massively changed after treatment as compared to pre-treated profiles. Nevertheless, we may assume that the genomic instability is not looked at the relevant resolution; nevertheless, it would be useful to further demonstrate the extent of this genome instability by whole genome sequencing before and after treatment, at some selected time points.

We thank the insightful comment from the reviewer. The link between increased genome instability and EZH2i has been indicated in previous studies^{18–22}. The preprint by Kazansly et al. further demonstrated that EZH2 inhibition induces newly acquired somatic mutations in various genes, including *RB1*, *EZH2*, *CHEK2*, *NF2*, *MAPK1*, and *EP300*, in SMARCB1-deficient sarcomas. The study also identified active transcription of *PGBD5*, a transposase-derived genes with retained nuclease activity, known to induce somatic mutations and double-stranded DNA breaks (DSBs). Indeed, they observed an induction of DSBs upon EZH2 inhibition, which corroborates our observation that EZH2 inhibition triggers DSBs in ATRT cancer cells (**Fig. 4a, b, e**). However, due to the high heterogeneity of cancer cell colonies and the trade-off between sequencing read length and accuracy, demonstrating genome instability through whole-genome sequencing (WGS) presents challenges, especially at the relevant resolution for DNA damage associated with L1-mediated reverse transcription. Although future investigations are warranted to map the extent of genomic instability induced by EZH2 inhibition in SMARCB1-deficient tumors, this particular aspect falls beyond the current scope of our manuscript.

5. it could be interesting to have a broader overview on how H3K27me3 is actually depleted in treated cells, not only at IR-Alu or selected derepressed gene regions, but at the whole genome level;

We thank the reviewer for the constructive suggestion to examine how H3K27me3 is depleted in the UNC1999-treated cells at the whole genome level. To address this, we re-analyzed H3K27me3 CUT&RUN data and observed a global depletion of the H3K27me3 signal in cells treated with UNC1999 versus UNC2400 at both day 4 and day 6 (New **Extended Data Fig. 3e**). This aligns with our expectations, as UNC1999 removes H3K27me3 marks across the genome.

Extended Data Fig. 3. e, Heatmap and average profile of the H3K27me3 CUT&RUN signal in samples treated with UNC1999 or UNC2400 at pooled peaks of the two conditions with ± 2 Kb up/downstream of the center of the peaks in UNC1999 versus UNC2400 day 4 (left) and day 6 (right).

6. How to explain then the reduced expression of intergenic IR-Alu? Is there any correlation with the depletion of H3K27me3 in intergenic regions versus intronic regions?

We appreciate the reviewer's comment and apologize for any confusion regarding the change in the expression of intergenic IR-Alu elements. The donut plots suggest an underrepresentation of intergenic IR-Alus in the upregulated IR-Alu repeats, as opposed to a reduction in expression, when compared to the expected genomic distribution (**Fig. 3a**). This can be explained by the H3k27me3 distribution analysis, indicating a larger fraction of baseline H3K27me3 peaks overlapping with intronic regions in the UNC2400-treated conditions (New **Extended Data Fig. 3h**). Given UNC1999's global depletion of H3K27me3 (New **Extended Data Fig. 3e**), we anticipate increased expression of intergenic IR-Alus. Indeed, our differential expression analysis revealed that more intergenic IR-Alus gained expression following UNC1999 treatment at both day 4 and day 6 (**Reviewer Figure 2**). In addition, a depletion of H3K27me3 was observed in both intergenic and intronic regions under UNC1999 treatment (**Reviewer Figure 3**), consistent with the removal of H3K27me3 across the genome.

Extended Data Fig. 3. h, Donut plots showing the distribution of the H3K27me3 peaks in samples treated with UNC2400 at day 4 (top) and day 6 (bottom). Odds ratio and p-value were calculated using the Fisher exact test by comparing the counts of the peaks at the genomic regions with the whole human genome annotation. * $p < 0.05$; *** $p < 0.001$; **** $p < 0.0001$.

Reviewer Figure 2. MA plots showing \log_2FC versus Average \log_2CPM for differential analysis of intergenic IR-Alus of UNC1999 versus UNC2400 at day 4 (left) and day 6 (right). Red and blue dots represent the upregulated and downregulated repeat elements respectively. Gray dots represent repeats that are not differentially regulated. Significance was determined by $|\log_2FC| > 1$, and $FDR < 0.05$.

Reviewer Figure 3. Heatmaps and average profile of the H3K27me3 CUT&RUN signal at intronic or intergenic regions in samples treated with UNC1999 or UNC2400, plotted with +/-2Kb upstream/downstream the centre of the intronic peaks in UNC1999 versus UNC2400 day 4 (left), intronic peaks in UNC1999 versus UNC2400 day 6 (second from left), intergenic peaks in UNC1999 versus UNC2400 day 4 (second from right) and intergenic peaks in UNC1999 versus UNC2400 day 6 (right).

7. The author insists on ISG among genes with co-induced expression of IR-Alu; is there any particular biological functions for all the other genes? What is the proportion of « non ISG » that show similar co-expression with IR-Alu and is that of particular interest to also look at those ones?

We thank the reviewer for the constructive suggestion to explore the biological functions of non-ISG genes co-induced with IR-Alus. We previously performed pathway analysis of the upregulated genes associated with UNC1999-induced IRs and uncovered that at day 6, the top significantly enriched pathways with highest precision score was interferon alpha/beta signaling, along with related pathways including viral defense response and innate immune response (**Fig. 3f, Extended Data Fig. 3f**). For other non-ISG genes co-induced with IR-Alus, we observed enrichment in multiple metabolism and biosynthesis processes (**Fig. 3f**). Additionally, by intersecting the genes co-induced with IR-Alus at day 4 or day 6 with the combined set of ISGs and IFN pathway genes, we noted a significant overrepresentation of the overlapping genes at day 6 compared to day 4 (New **Extended Data Fig. 4h**). This supports the enrichment of ISGs among the upregulated genes associated with UNC1999-induced IR-Alus. The pronounced enrichment of ISGs is particularly intriguing, as this may provide a feedforward mechanism for enhancing viral mimicry. In contrast, the proportion of non-ISG genes co-induced with IR-Alus decreased from 87.5% (49 out of 56) at day 4 to 61.6% (45 out of 73) at day 6 (New **Extended Data Fig. 4h**). Although the non-ISG genes co-induced with IR-Alus show enrichment in metabolism and biosynthesis pathways, a detailed investigation of their biological relevance falls beyond the scope of our current manuscript.

Figure 3. f, Pathway analysis of upregulated genes ($n=207$) that overlap with upregulated intragenic IRs in UNC1999 versus UNC2400 at day 6. P-value is calculated using the hypergeometric test followed by correction for multiple testing.

Extended Data Fig. 4. h, Venn diagram showing the intersection between the union set of the ISG genes and IFN pathway genes with the set of upregulated genes that overlap with upregulated IR-Alus in UNC1999 versus UNC2400-treated samples at day 4 (top row) and day 6 (bottom row). The count of upregulated ISG genes overlapping with upregulated IR Alus at day 6 was compared to that at day 4 using the one-tailed Fisher exact test to calculate the odds ratio and p-value.

Reviewer #3 (Remarks to the Author):

Inhibiting epigenetic regulators in tumors has the potential to activate a viral mimicry state, which stimulates antitumoral innate immunity. However, each tumor type can depend on different pathways and be vulnerable to a distinct set of epigenetic therapies. Here, the authors explore the mechanisms underlying the vulnerability of tumors deficient for the SWI/SNF complex (SMARCB1 loss-of-function) upon EZH2 inhibition, using atypical teratoid rhabdoid tumors (ATRTs) as a model. Their data confirm that EZH2i leads to a viral mimicry state as shown in other tumor types and demonstrate that this process results from the dual activation of cytosolic RNA and DNA sensing pathways. Finally, some experiments suggest that retrotransposons (Alu/L1) may trigger the activation of these pathways through different mechanisms. The study tackles fundamental questions to fully understand the mechanism of action of epigenetic drugs and will help understanding the role of endogenous retroelements in immunity. However, the link between Alu repeats and RNA sensing, as well as between L1 and DNA sensing, is preliminary and needs to be reinforced before publication.

Pros:

- new findings revealing the pathways underlying the activation of viral mimicry in tumor cells
- strong demonstration that both dsRNA and cytoplasmic DNA sensing cooperate to set up viral mimicry upon dual inactivation of EZH2 and SMARCB1.

Cons:

- the causal relationship between Alu/L1 and viral mimicry is not proven in this context

We thank reviewer #3 for the positive comments and the constructive criticism. We have addressed each comment individually below. These suggestions have greatly improved our revised manuscript.

Major points:

1. The proposed model is that Alu inverted repeats embedded in genes derepressed by EZH2i may generate structured RNA bound by MDA5, which in turn will activate MAVS and the dsRNA sensing pathway. However, the authors do not directly show here that Alu-IR are actually bound by MDA5 in the context of ATRTs. They could perform an MDA5 protection sequencing assay as achieved by the same group for demethylating agents (Mehdipour et al. Nature 2020). Also, do dsRNA foci detected by IF contain Alu sequences by RNA-FISH?

We thank the reviewer for the constructive suggestion to further validate that intragenic IR-Alus can activate the dsRNA sensing pathway. To address this, we immunoprecipitated dsRNA from total RNA in BT16 cells using the J2 antibody and examined the enrichment of Alu elements in

dsRNA. The qRT-PCR analysis targeting Alu consensus sequences revealed significantly higher Alu enrichment in the dsRNA of UNC1999-treated cells (New **Extended Data Fig. 3d**). Additionally, using primers designed for experimentally validated (EV) immunogenic IR-Alus that gained expression following EZH2 inhibition, we observed higher enrichment in dsRNA from UNC1999-treated cells compared to UNC2400 (New **Fig. 2h**). Furthermore, primers targeting representative IR-Alus that gained expression post-EZH2 inhibition revealed a majority of these IR-Alu pairs to be significantly more abundant in the dsRNA species of UNC1999-treated cells (New **Fig. 2i**). These findings collectively suggest that IR-Alus induced by EZH2 inhibition can form dsRNA, potentially activating the dsRNA sensing pathway.

Extended Data Fig. 3. d, Error bar plots showing the enrichment of transcripts of Alu repeats or GAPDH in the dsRNA species immunoprecipitated with J2 antibody from total RNA harvested from UNC1999- or UNC2400-treated BT16 cells. qRT-PCR was employed for analysis, with normalization to the corresponding input RNA. Error bars represent the SD of three independent experiments. *** $p < 0.001$ (two-tailed t test).

Fig. 2. h,i, Error bar plots showing the enrichment of indicated transcripts of upregulated experimentally validated (EV) IR-Alus (**h**) or annotated IR-Alus (**i**) in the dsRNA species immunoprecipitated with J2 antibody from total RNA harvested from UNC1999- or UNC2400-treated BT16 cells. qRT-PCR was employed for analysis, with normalization to the corresponding input RNA. Error bars represent the SD of three independent experiments. * $p < 0.05$; ** $p < 0.01$ (two-tailed t test).

2. The link between the so-called L1 retrotransposition and cytoplasmic DNA sensing is weak in many respects.

2a. First, it appears that the expressed L1s belong to families which are unable to retrotranspose (not L1Hs) and it is unclear whether they are transcribed autonomously (from the L1 promoter) or “passively” (cotranscribed with genes in which they are embedded). A careful presentation and analysis of the full length elements assumed to be expressed is needed here. ORF1p western-blots may also reinforce the hypothesis that some intact and young L1s are expressed. Similarly, what is the extent of overlap between H3K27me3 reduction, STING activation and ORF1p expression at the single-cell level (IF)?

We thank the reviewer for the constructive suggestion to further assess the expression of full-length L1s. To address the reviewer’s comment, we conducted western blotting and immunostaining, revealing upregulation in the expression of L1 ORF1p under UNC1999-treated conditions (New **Extended Data Fig. 7g, h**). This reinforces our hypothesis that a subset of intact and young L1s actively expresses and may possess reverse transcriptase activity upon EZH2 inhibition. Furthermore, through re-analysis of CUT&RUN and RNA-seq data at both day 4 and day 6, we observed H3K4me3 enrichment at the promoter regions of upregulated full-length L1s. This was accompanied by a depletion of H3K27me3 and an increase in RNA-seq signal, suggesting active transcription of these L1s post UNC1999 treatment (New **Extended Data Fig. 7f**). Notably, in response to Reviewer #3 comment 2b, we demonstrated the dependence of the formation of cytoplasmic ssDNA and activation of the DNA sensing pathway on L1 activity. Collectively, these results validate that a specific group of full-length L1s, derepressed by EZH2 inhibition, undergo active expression, potentially serving as a crucial source of enzymatic machinery for reverse transcription.

In investigating the relationship among H3K27me3 reduction, LINE-1 ORF1p expression, and STING activation, we conducted immunostaining using antibodies against H3K27me3, STING, and LINE-1 ORF1p in BT16 cells. Following UNC1999 treatment, a depletion of H3K27me3 staining was observed in nearly all cells, with the majority showing increased expression of both STING and LINE-1 ORF1p (New **Fig. 5a, b**). These results strongly indicate a connection between H3K27me3 reduction, enhanced full-length LINE-1 expression, and the activation of cytoplasmic DNA sensing.

Extended Data Fig. 7. g, Representative immunoblots showing the expression level of L1 ORF1p (left) and the quantification of relative signal intensity of L1 ORF1p using ImageJ (right) in BT16 cells treated with either UNC2400 or UNC1999. Vinculin was used as a loading control. Error bars represent the SD of two independent experiments.

Extended Data Fig. 7. h, Confocal microscopy of anti-L1 ORF1p immunofluorescence in BT16 cells treated with either UNC2400 or UNC1999. L1 ORF1p was stained in red, and nuclei were stained in blue (DAPI). Scale bars, 10 μ m.

Extended Data Fig. 7. f, Genome track signal of the RNA-seq and of H3K4me3 and H3K27me3 CUT&RUN marks at representative upregulated full-length LINE-1 elements from samples treated with UNC1999 or UNC2400.

Fig. 5. a, Confocal microscopy of anti-H3K27me3 and anti-L1 ORF1p immunofluorescence in BT16 cells treated with either UNC2400 or UNC1999. H3K27me3 was stained in green, L1 ORF1p was stained in red, and nuclei were stained in blue (DAPI). Scale bars, 10 μ m.

Fig. 5. b, Confocal microscopy of anti-STING and anti-L1 ORF1p immunofluorescence in BT16 cells treated with either UNC2400 or UNC1999. STING was stained in green, L1 ORF1p was stained in red, and nuclei were stained in blue (DAPI). Scale bars, 10 μ m.

2b. Second, the detection of ssDNA in the cytoplasm is not evidence of retrotransposition. The authors detect cytoplasmic ssDNA upon EZH2i, and these DNA species disappear after treatment with drugs known to inhibit ORF2p reverse transcriptase activity. Nevertheless, these drugs may affect other polymerases and it still is unknown whether the ssDNA species contain L1 sequences or depend on L1 activity. FISH using L1 probes and L1 knock-down could strengthen this conclusion (see De Cecco et al. Nature 2019, Thomas et al. Cell Stem Cell 2017 for related experiments).

We thank the reviewer for the constructive criticism. To investigate whether the ssDNA species contain LINE-1 elements, we obtained ssDNA species by extracting extrachromosomal DNA followed by digestion with dsDNase²³ and analyzed LINE-1 enrichment using qRT-PCR. We found that LINE-1 elements were significantly more abundant in UNC1999-treated samples (New Fig. 5c), suggesting that ssDNA species induced upon EZH2 inhibition are enriched for LINE-1 elements.

Fig. 5. c, Error bar plots showing the enrichment of indicated regions of LINE-1 elements in the ssDNA species extracted from UNC1999- or UNC2400-treated BT16 cells. qRT-PCR was employed for analysis, with normalization to the corresponding gDNA. Error bars represent the SD of three independent experiments. * $p < 0.05$; ** $p < 0.01$; *** $p < 0.001$ (two-tailed t test).

We further assessed the dependence of ssDNA induction on LINE-1 activity by generating LINE-1-deficient cells using sgRNAs targeting the 5'UTR, ORF1, or ORF2 of L1 (New **Extended Data Fig. 7i**). Our findings revealed a significant attenuation in the induction of ssDNA species in the LINE-1-deficient cells upon EZH2 inhibition (New **Fig. 5d**). Consequently, we found that the activation of ISGs following EZH2 inhibition was significantly abolished with the inactivation of the LINE-1 elements (New **Fig. 5e**). Recently, two significant works by Thawani Nature 2023²⁴ and Baldwin Nature 2023²⁵ elucidated the structural basis and molecular mechanism underlying the LINE-1 retrotransposon. They also revealed that cytosolic ORF2p RT activity can generate RNA:DNA hybrids, which triggers the activation of cGAS/STING-mediated DNA sensing and downstream interferon signaling. These findings align with our discovery that cytoplasmic ssDNA induced by UNC1999 treatment relies on the reverse transcriptase activity of LINE-1. Together, these data suggest that LINE-1-mediated reverse transcription may contribute to the presence of cytoplasmic ssDNA and subsequent activation of DNA sensing following EZH2 inhibition.

Extended Data Fig. 7. i, The expression of L1 elements in BT16 cell line with sgRNAs against negative control, L1-5'UTR, L1-ORF1, or L1-ORF2 was measured by quantitative real-time PCR. Error bars represent the SD of three independent experiments. * $p < 0.05$; ** $p < 0.01$; *** $p < 0.001$ (two-tailed t test).

Fig. 5. d, Confocal microscopy of anti-ssDNA immunofluorescence in BT16 cell line with sgRNAs against negative control, L1-5'UTR, L1-ORF1, or L1-ORF2 at day 4 post the UNC1999 or UNC2400 treatment. ssDNA was stained in red, and nuclei were stained in blue (DAPI). Scale bars, 10 μ m.

Fig. 5. e, The expression of four indicated interferon-responsive genes in BT16 cell line with sgRNAs against negative control, L1-5'UTR (top row), L1-ORF1 (middle row), or L1-ORF2 (bottom row) was measured by quantitative real-time PCR at day 6 post the UNC1999 or UNC2400 treatment. Error bars represent the SD of three independent experiments. ** $p < 0.01$; *** $p < 0.001$ (two-tailed t test).

2c. Finally, the term “retrotransposition” is misemployed. “L1 retrotransposition” refers to the entire cycle of L1 replication, which includes expression, assembly of L1 RNPs, and target-primed reverse transcription, ultimately leading to new insertions. I would suggest the authors substitute this term by “L1 reverse transcription” or “L1 enzymatic activities” in the context of their experiments.

We thank the reviewer for this comment. We have substituted the term “L1 retrotransposition” by “L1-mediated reverse transcription” in the revised manuscript to avoid confusion and to improve accuracy.

Minor points:

3. What is the % of H3K27me3-bound Alu-IR that are upregulated, and how does it compare to unbound Alu-IR ?

We thank the reviewer for this comment. In response, we have generated plots depicting the percentage of upregulated IR-Alus overlapping with baseline H3K27me3 peaks. At day 4, 13.1% of upregulated IR-Alus (81 out of 617) intersect with H3K27me3 peaks, while at day 6, 15.2% (192 out of 1260) exhibit overlap (**Reviewer Figure 4**). Moreover, we intersected the promoters of genes associated with upregulated IR-Alu pairs with the baseline H3k27me3 peaks. Remarkably, 76.5% of IR-pair-associated genes (315 out of 412) overlap with H3k27me3 peaks at day 4, and at day 6, 74.3% of upregulated IR-Alus (641 out of 863) exhibit overlap (**Reviewer Figure 4**). These findings collectively suggest that H3K27me3 preferentially regulates IR-Alus by targeting their associated genes.

Reviewer Figure 4. Stacked bar plots depicting the percentage of upregulated IR-Alus induced by UNC1999 (left) and the genes associated with upregulated IR-Alu pairs (right) overlapping with baseline H3K27me3 peaks at day 4 and day 6.

4. Refs 4 and 18 appear to be duplicates.

We thank the reviewer for bringing this to our attention. We have corrected the duplication of Refs 4 and 18 in the manuscript.

References:

1. Ribas, A. *et al.* Oncolytic Virotherapy Promotes Intratumoral T Cell Infiltration and Improves Anti-PD-1 Immunotherapy. *Cell* **170**, 1109-1119.e10 (2017).
2. Ribas, A. *et al.* SD-101 in Combination with Pembrolizumab in Advanced Melanoma: Results of a Phase Ib, Multicenter Study. *Cancer Discov* **8**, 1250–1257 (2018).
3. Ishizuka, J. J. *et al.* Loss of ADAR1 in tumours overcomes resistance to immune checkpoint blockade. *Nature* **565**, 43–48 (2019).
4. Demaria, O. *et al.* Harnessing innate immunity in cancer therapy. *Nature* **574**, 45–56 (2019).
5. Chen, R., Ishak, C. A. & De Carvalho, D. D. Endogenous retroelements and the viral mimicry response in cancer therapy and cellular homeostasis. *Cancer Discov* **11**, 2707–2725 (2021).
6. Jones, P. A., Ohtani, H., Chakravarthy, A. & De Carvalho, D. D. Epigenetic therapy in immune-oncology. *Nature Reviews Cancer* vol. 19 151–161 Preprint at <https://doi.org/10.1038/s41568-019-0109-9> (2019).
7. Snell, L. M., McGaha, T. L. & Brooks, D. G. Type I Interferon in Chronic Virus Infection and Cancer. *Trends Immunol* **38**, 542–557 (2017).
8. Roulois, D. *et al.* DNA-Demethylating Agents Target Colorectal Cancer Cells by Inducing Viral Mimicry by Endogenous Transcripts. *Cell* (2015) doi:10.1016/j.cell.2015.07.056.
9. Chiappinelli, K. B. *et al.* Inhibiting DNA Methylation Causes an Interferon Response in Cancer via dsRNA Including Endogenous Retroviruses. *Cell* (2015) doi:10.1016/j.cell.2015.07.011.
10. Deblois, G. *et al.* Epigenetic switch–induced viral mimicry evasion in chemotherapy-resistant breast cancer. *Cancer Discov* **10**, 1312–1329 (2020).
11. Morel, K. L. *et al.* EZH2 inhibition activates a dsRNA–STING–interferon stress axis that potentiates response to PD-1 checkpoint blockade in prostate cancer. *Nat Cancer* **2**, 444–456 (2021).
12. Sheng, W. *et al.* LSD1 Ablation Stimulates Anti-tumor Immunity and Enables Checkpoint Blockade. *Cell* **174**, 549-563.e19 (2018).
13. Cañadas, I. *et al.* Tumor innate immunity primed by specific interferon-stimulated endogenous retroviruses. *Nat Med* (2018) doi:10.1038/s41591-018-0116-5.
14. Shen, J. Z. *et al.* FBXO44 promotes DNA replication-coupled repetitive element silencing in cancer cells. *Cell* **184**, 352-369.e23 (2021).
15. Leruste, A. *et al.* Clonally Expanded T Cells Reveal Immunogenicity of Rhabdoid Tumors. *Cancer Cell* **36**, 597–612 (2019).
16. Mehdipour, P. *et al.* Epigenetic therapy induces transcription of inverted SINEs and ADAR1 dependency. *Nature* **588**, 169–173 (2020).
17. Liu, H. *et al.* Tumor-derived IFN triggers chronic pathway agonism and sensitivity to ADAR loss. *Nat Med* **25**, 95–102 (2019).
18. Liao, Y. *et al.* Inhibition of EZH2 transactivation function sensitizes solid tumors to genotoxic stress. *Proc Natl Acad Sci U S A* **119**, (2022).
19. Xu, L. *et al.* Pharmacological inhibition of EZH2 combined with DNA-damaging agents interferes with the DNA damage response in MM cells. *Mol Med Rep* **49**, 4249–4255 (2019).

20. Wang, Y. *et al.* DNA-PK-mediated phosphorylation of EZH2 regulates the DNA damage-induced apoptosis to maintain T-cell genomic integrity. *Cell Death Dis* **7**, e2316–e2316 (2016).
21. Campbell, S., Ismail, I. H., Young, L. C., Poirier, G. G. & Hendzel, M. J. Polycomb repressive complex 2 contributes to DNA double-strand break repair. *Cell Cycle* **12**, 2675–2683 (2013).
22. Wu, Z. *et al.* Polycomb protein EZH2 regulates cancer cell fate decision in response to DNA damage. *Cell Death Differ* **18**, 1771–1779 (2011).
23. Thomas, C. A. *et al.* Modeling of TREX1-Dependent Autoimmune Disease using Human Stem Cells Highlights L1 Accumulation as a Source of Neuroinflammation. *Cell Stem Cell* **21**, 319-331.e8 (2017).
24. Thawani, A., Ariza, A. J. F., Nogales, E. & Collins, K. Template and target site recognition by human LINE-1 in retrotransposition. *Nature* (2023) doi:10.1038/s41586-023-06933-5.
25. Baldwin, E. T. *et al.* Structures, functions, and adaptations of the human LINE-1 ORF2 protein. *Nature* (2023) doi:10.1038/s41586-023-06947-z.

REVIEWER COMMENTS

Reviewer #2 (Remarks to the Author):

The revised manuscript by Feng et al addresses one by one each of the issues that was raised, by myself and by other reviewers, in a precise and quite comprehensive way. All results that are provided confirmed the ones that were presented in the previously submitted manuscript. In details,

the authors have added a shRNA knockdown of EZH2 in addition to the pharmacological inhibition. The expected effects are indeed observed both on cell viability and on interferon signaling gene expression.

2) the authors have extended the treatment experiments up to 18 days. Although it is still clearly much less than what is used in the clinics, this additional experiments confirm that longer exposure is still associated with a similar induction of interferon signaling gene expression, in a range that is decreasing over time but remains highly significant at late time points.

3) the authors have reanalyses existing data to demonstrate that before any EZH2 inhibition, the viral mimicry process is already active in SMARCB1 deficient tissues more than in SMARCB1 proficient tissues; IR-Alu in particular are influenced by SMARCB1 re-expression. The authors add to this an « ISG score » correlated to the expression of SMARCB1 which altogether provides the referent levels of viral mimicry activity that was lacking in the previous version.

4) the authors don't really address the question of the actual impact of genomic instability; at present, the assumption that viral mimicry induces DSBs is not corroborated by the identification of remaining scars of such DSBs in the genome. Whole genome sequencing may not easily demonstrate this genomic instability at the relevant resolution, but this should at least be discussed as a perspective.

5) the dramatic decrease of H3K27me3 marks upon treatment is definitely convincing.

6) the confusion is clarified.

7) the answer is satisfying.

Altogether, I consider that the authors correctly answers the questions that were raised, and that the manuscript could be published as it stands.

Reviewer #3 (Remarks to the Author):

In response to my initial concerns regarding the triggers of viral mimicry following EZH2i treatment, the authors have conducted new experiments. These experiments convincingly demonstrate a correlation between viral mimicry induction and the presence of double-stranded RNA structures containing Alu sequences. Additionally, they strengthen the association between viral mimicry and both L1 expression and L1 ssDNA accumulation. However, further clarification is needed regarding the direct causal role of L1s in inducing viral mimicry.

A key new experiment in this revised manuscript is CRISPR-mediated knock out/down of L1 elements, which leads to a drastic reduction of the interferon response upon EZH2 inhibition. In principle, this experiment should formally prove that viral mimicry depends on L1 expression, consistent with the effects of RT inhibitors. However, the interpretation of results becomes challenging based on the provided controls. Three pairs of guide RNAs were designed against L1 5'UTR (which contains the L1 promoter), L1 ORF1 and L1 ORF2, respectively. While qRT-PCR shows an approximate 50% decrease in L1 expression (Ext Data 7i), the observed reduction in L1 expression is unexpected because Cas9-mediated cleavage in ORFs is expected to generate indels, leading to mutated L1 RNA species that could still be transcribed. Moreover, heterogeneity in mutated cell populations would be expected in ssDNA IF experiments. Finally, it is unclear how

many copies and which families can be targeted by the gRNAs. Overall, the characterization of Cas9-mediated inactivation of L1s seems incomplete and preliminary. The authors should provide a clearer understanding of the specific targets and how L1s are affected by this manipulation. Concerns could also be raised about potential massive chromosomal rearrangements.

l. 419-421 and Ext Fig. 7f: In the examples provided, full-length L1 transcription appears to be pervasive (not restricted to the boundaries of the L1 element) and H3K4me3 enrichment does not show a distinct peak within L1 promoter sequence, as would be expected for an actively transcribed L1 element. Furthermore, they represent relatively ancient L1PA elements, which may not possess functional reverse transcriptase activity. Therefore, caution is warranted when interpreting these findings in the context of L1-mediated viral mimicry. Please add a caveat.

Other minor points:

- ORF1p antibody accession is incorrect.
- l.372: duplicate "has been"

REVIEWER COMMENTS

Reviewer #2 (Remarks to the Author):

The revised manuscript by Feng et al addresses one by one each of the issues that was raised, by myself and by other reviewers, in a precise and quite comprehensive way. All results that are provided confirmed the ones that were presented in the previously submitted manuscript. In details, the authors have added a shRNA knockdown of EZH2 in addition to the pharmacological inhibition. The expected effects are indeed observed both on cell viability and on interferon signaling gene expression.

2) the authors have extended the treatment experiments up to 18 days. Although it is still clearly much less than what is used in the clinics, this additional experiments confirm that longer exposure is still associated with a similar induction of interferon signaling gene expression, in a range that is decreasing over time but remains highly significant at late time points.

3) the authors have reanalyses existing data to demonstrate that before any EZH2 inhibition, the viral mimicry process is already active in SMARCB1 deficient tissues more than in SMARCB1 proficient tissues; IR-Alu in particular are influenced by SMARCB1 re-expression. The authors add to this an « ISG score » correlated to the expression of SMARCB1 which altogether provides the referent levels of viral mimicry activity that was lacking in the previous version.

4) the authors don't really address the question of the actual impact of genomic instability; at present, the assumption that viral mimicry induces DSBs is not corroborated by the identification of remaining scars of such DSBs in the genome. Whole genome sequencing may not easily demonstrate this genomic instability at the relevant resolution, but this should at least be discussed as a perspective.

We appreciate the reviewer highlighting the need to address the actual impact of genomic instability. Due to the high heterogeneity of cancer cell colonies and the trade-off between sequencing read length and accuracy, demonstrating genome instability through whole-genome sequencing (WGS) presents challenges, especially at the resolution necessary for detecting DNA damage associated with L1-mediated reverse transcription. As the reviewer suggested, we have discussed the need to further elucidate the impact of genomic instability in our manuscript. Below is the included perspective:

“In our study, we observed that inhibiting EZH2 leads to the induction of DSBs, as evidenced by increased γ -H2AX foci formation, which could be rescued with the addition of RT inhibitors. However, the direct evidence of remaining genomic scars of DSB induction is lacking. This limitation stems from the challenges posed by WGS in capturing genomic instability at the required resolution. Future studies should employ higher-resolution techniques, such as long-read sequencing or single-cell sequencing, to identify potential DSBs and their remnants. These

methods could complement our findings and offer a more comprehensive understanding of the impact of EZH2i on L1 activity and genome instability."

5) the dramatic decrease of H3K27me3 marks upon treatment is definitely convincing.

6) the confusion is clarified.

7) the answer is satisfying.

Altogether, I consider that the authors correctly answers the questions that were raised, and that the manuscript could be published as it stands.

We appreciate the reviewer's thorough review and positive feedback, which have greatly improved our manuscript. We are delighted that the manuscript meets the reviewer's expectations for publication.

Reviewer #3 (Remarks to the Author):

In response to my initial concerns regarding the triggers of viral mimicry following EZH2i treatment, the authors have conducted new experiments. These experiments convincingly demonstrate a correlation between viral mimicry induction and the presence of double-stranded RNA structures containing Alu sequences. Additionally, they strengthen the association between viral mimicry and both L1 expression and L1 ssDNA accumulation. However, further clarification is needed regarding the direct causal role of L1s in inducing viral mimicry.

A key new experiment in this revised manuscript is CRISPR-mediated knock out/down of L1 elements, which leads to a drastic reduction of the interferon response upon EZH2 inhibition. In principle, this experiment should formally prove that viral mimicry depends on L1 expression, consistent with the effects of RT inhibitors. However, the interpretation of results becomes challenging based on the provided controls. Three pairs of guide RNAs were designed against L1 5'UTR (which contains the L1 promoter), L1 ORF1 and L1 ORF2, respectively. While qRT-PCR shows an approximate 50% decrease in L1 expression (Ext Data 7i), the observed reduction in L1 expression is unexpected because Cas9-mediated cleavage in ORFs is expected to generate indels, leading to mutated L1 RNA species that could still be transcribed. Moreover, heterogeneity in mutated cell populations would be expected in ssDNA IF experiments. Finally, it is unclear how many copies and which families can be targeted by the gRNAs. Overall, the characterization of Cas9-mediated inactivation of L1s seems incomplete and preliminary. The authors should provide a clearer understanding of the specific targets and how L1s are affected by this manipulation. Concerns could also be raised about potential massive chromosomal rearrangements.

We thank the reviewer for the valuable feedback, which has significantly improved the robustness of our findings.

CRISPR-Cas9 targeting of ORFs can induce indels in the coding sequences, causing frameshift mutations that introduce premature stop codons. The resulting mRNA either produces a truncated, nonfunctional protein or is degraded by nonsense-mediated decay (NMD), leading to a reduction in protein levels and, in many cases, a decrease in mRNA transcripts¹. On the other hand, indels in the promoter region can disrupt transcription factor binding sites, leading to decreased transcription of the gene^{2,3}. In addition, the disruption of LINE1 protein following CRISPR knockout likely destabilizes the ribonucleoprotein (RNP) complex formed by LINE1 mRNA, ORF1p, and ORF2p⁴, leading to a decrease in LINE1 transcripts. These mechanisms likely explain the observed decrease in L1 transcription (Previous **Extended Data Fig. 7i**, New **Extended Data Fig. 8e**).

Our revised manuscript now includes a more detailed analysis of the specific L1 copies and families targeted by our gRNAs designed against L1 5'UTR⁵ or ORF1. Using bioinformatics tools, we predicted the specific L1 loci affected by CRISPR. Specifically, we performed Smith-Waterman pairwise local alignment of full-length L1PA2-L1PA7 elements to L1H (New **Extended Data Fig. 8a**). The plots display tracks of these alignments, highlighting guide RNA locations, gaps, mismatches, and matches. The top plot shows the first 900 bp of the sequence alignment, including the 5'UTR guide markings, while the bottom plot shows the second 1KB of the alignment with ORF1 guide markings. Additionally, we generated donut plots depicting the counts of LINE-1 elements in the genome that are aligned with either 5'UTR guide and ORF1 guide (New **Extended Data Fig. 8b**). These data suggest that the gRNA-targeted sequences are conserved across multiple L1 families including L1H, L1PA2 and L1PA3, with a higher specificity to L1H.

To further substantiate our findings, we performed IF analyses, which demonstrated a significant decrease in LINE1 ORF1p levels following CRISPR KO using gRNAs against L1 5'UTR or ORF1 (New **Extended Data Fig. 8c, d**).

We apologize for not fully representing the variability in ssDNA puncta per cell in the CRISPR-edited cell populations with our initial IF images. To address these concerns, we have quantified the number of ssDNA puncta per cell across different samples, using a sample size of 150 cells as a consistent benchmark across all conditions (New **Extended Data Fig. 5e**). As expected, this quantification indeed highlights the observed heterogeneity in ssDNA IF experiments. Accordingly, we have replaced some of the IF images with more representative ones, providing a clearer depiction of the variability in the response

Together, these enhancements and clarifications strengthen our results using CRISPR-mediated knock out/down of L1 elements and provide a more comprehensive understanding of the mechanisms underlying viral mimicry induced by EZH2 inhibition.

l. 419-421 and Ext Fig. 7f: In the examples provided, full-length L1 transcription appears to be pervasive (not restricted to the boundaries of the L1 element) and H3K4me3 enrichment does not show a distinct peak within L1 promoter sequence, as would be expected for an actively transcribed L1 element. Furthermore, they represent relatively ancient L1PA elements, which may not possess functional reverse transcriptase activity. Therefore, caution is warranted when interpreting these findings in the context of L1-mediated viral mimicry. Please add a caveat.

We appreciate the reviewer's observation. As suggested, we have included the following caveat:

“Our reverse transcriptase inhibition and L1 knockout/down experiments suggest that the presence of cytoplasmic ssDNA and subsequent DNA sensing activation following EZH2i depend on L1 reverse transcriptase activity. Recent studies by Thawani et al.⁶ and Baldwin et al.⁷ provided comprehensive insights into the structural basis and molecular mechanism of the L1 retrotransposon. They demonstrated that the cytosolic reverse transcriptase activity of ORF2p can generate RNA:DNA hybrids, leading to the activation of DNA sensing pathway and downstream interferon signaling. These findings align with our discovery that cytoplasmic ssDNA induced by UNC1999 relies on the reverse transcriptase activity of L1 elements. However, our CUT&RUN and RNA-seq analyses did not pinpoint the specific L1 subfamily involved and might not have the resolution to detect autonomous L1Hs that could potentially be activated by EZH2i. The relatively ancient L1PA elements upregulated by EZH2i may lack the functional reverse transcriptase activity necessary for retrotransposition. It is conceivable that L1PA elements upregulated by EZH2i could exploit the reverse transcriptase produced by L1Hs expressed at basal levels to mobilize in a nonautonomous manner. These limitations should be considered when extrapolating our results to active L1 elements involved in viral mimicry.”

Other minor points:

- ORF1p antibody accession is incorrect.

We thank the reviewer for bringing this to our attention. We have corrected the accession of the ORF1p antibody in our manuscript.

- l.372: duplicate “has been”

We thank the reviewer for bringing this to our attention. We have corrected the duplication in the manuscript.

References:

1. Tuladhar, R. *et al.* CRISPR-Cas9-based mutagenesis frequently provokes on-target mRNA misregulation. *Nat Commun* **10**, 4056 (2019).
2. Akinci, E., Hamilton, M. C., Khowpinitchai, B. & Sherwood, R. I. Using CRISPR to understand and manipulate gene regulation. *Development* **148**, dev182667 (2021).
3. Diao, Y. *et al.* A tiling-deletion-based genetic screen for cis-regulatory element identification in mammalian cells. *Nat Methods* **14**, 629–635 (2017).
4. Mita, P. *et al.* LINE-1 protein localization and functional dynamics during the cell cycle. *Elife* **7**, e30058 (2018).
5. Briggs, E. M. *et al.* Unbiased proteomic mapping of the LINE-1 promoter using CRISPR Cas9. *Mob DNA* **12**, 21 (2021).
6. Thawani, A., Ariza, A. J. F., Nogales, E. & Collins, K. Template and target site recognition by human LINE-1 in retrotransposition. *Nature* (2023) doi:10.1038/s41586-023-06933-5.
7. Baldwin, E. T. *et al.* Structures, functions, and adaptations of the human LINE-1 ORF2 protein. *Nature* (2023) doi:10.1038/s41586-023-06947-z.

REVIEWERS' COMMENTS

Reviewer #4 (Remarks to the Author):

The authors have addressed all of my comments in full and I have no reservations in recommending publication.

Reviewer #4 (Remarks on code availability):

The provided link does not allow access to embargoed code ([https://zenodo.org/records/10534911?token=eyJhbGciOiJIUzUxMiIsImIhdCI6MTcwNTY4NDQx\[...\]-xy8XoSNmZSG-Awpe7YexG3sjcTW6O4nNRy9h4WOrxE0hRI95kNtXCVG7gX4mw](https://zenodo.org/records/10534911?token=eyJhbGciOiJIUzUxMiIsImIhdCI6MTcwNTY4NDQx[...]-xy8XoSNmZSG-Awpe7YexG3sjcTW6O4nNRy9h4WOrxE0hRI95kNtXCVG7gX4mw)). Code is embargoed until January 2026, but should be made publicly available before publication.